

# Sensitivity of centennial mass loss projections of the Amundsen basin to the friction law

Julien Brondex[1], Fabien Gillet-Chaulet[1], and Olivier Gagliardini[1]

[1]Univ. Grenoble Alpes, CNRS, IRD, IGE, F-38000 Grenoble, France

**Correspondence:** J. Brondex (julien.brondex@gadz.org)

**Abstract.** Reliable projections of ice sheets future contribution to sea level rise require models able to accurately simulate grounding line dynamics, starting from initial states consistent with observations. Here, we simulate the centennial evolution of the Amundsen Sea Embayement in response to a schematic perturbation in order to assess the sensitivity of mass loss projections to the chosen friction law, depending on the initialisation strategy. To this end, three different model states are

constructed by inferring both the initial basal shear stress and viscosity fields with various relative weights. Then, starting from each of these model states, prognostic simulations are carried out using a Weertman, a Schoof and a Budd friction law, with different parameter values. Independently of the considered model state, the Weertman law systematically predicts the lowest mass losses. Although the sensitivity of projections to the chosen friction law tends to decrease when more weight is put on viscosity during initialisation, it remains significant for the most physically acceptable of the constructed model states.

In addition, because of its particular dependence on effective pressure, the Budd friction law induces significantly different GL retreat patterns than the other laws and predicts much higher mass losses.

## 1  Introduction

The West Antarctic Ice Sheet mean annual contribution to global sea level rise (SLR) has tripled over the last 25 years as a consequence of a growing imbalance between the mass it receives as snowfall and that which is discharged to the oceans by ice

streams (Shepherd et al., 2018). The most active basin of this region is the Amundsen Sea Embayment (ASE) where marine-terminating outlet glaciers draining ice to the oceans have shown sustained acceleration and thinning over the last decades, with their grounding lines (GL), i.e. the limit between the grounded ice-sheet and the floating ice shelf, retreating at rates higher than $1 \text{ km a}^{-1}$ since 1992 (Mouginot et al., 2014; Rignot et al., 2014). These observations raise concern regarding the near future of the ASE as they suggest that this sector of West Antarctica is undergoing a marine ice-sheet instability (MISI) which

would imply a significant additional global SLR in the coming decades (Joughin et al., 2014; Favier et al., 2014; Cornford et al., 2015). Producing trustworthy estimates of this future contribution requires accurate modelling of the GL dynamics on subcentennial timescales as well as the ability to produce model states as close as possible to the observations.

Using a synthetic flowline geometry, Brondex et al. (2017) have shown that the GL dynamics depends critically on the choice of the friction law, i.e. the mathematical relationship linking basal shear stress to other parameters including sliding

velocity. The ice/bed interface being usually out of reach, the formulation of a friction law has been a long standing problem



in glaciology. Various laws intended to describe different physical processes at the roots of basal motion have come up over the years (Weertman, 1957; Budd et al., 1979; Schoof, 2005; Tsai et al., 2015). Most ice flow modelling studies published so far use the Weertman friction law, which aims at describing the basal motion of ice over and around obstacles of a rigid bedrock by a combination of viscous creep and regelation (Weertman, 1957). However, many rapid ice streams of Antarctica

are known to be lying on soft beds made of water-laden till, the deformation of which explains most of the motion observed at the surface (Blankenship et al., 1986; Alley et al., 1986; Kavanaugh and Clarke, 2006). Numerous laboratory studies on till samples and in situ measurements have shown that at large strain the till rheology is plastic with a critical strength $\tau^*$ depending on effective pressure $N$, i.e. the difference between ice overburden pressure and water pressure (Boulton and Jones, 1979; Blankenship et al., 1986; Alley et al., 1986). To account for both the cases of rigid and soft beds, Tsai et al. (2015)

proposed a law inducing a Coulomb friction regime at low $N$ which instantaneously switches to a Weertman friction regime at higher $N$. By construction, this law induces an upper bound, function of $N$, of the basal shear stress. Although it was originally intended to describe the ice flowing over a rigid bed with the opening of water-filled cavities, the law proposed by Schoof (2005) behaves very similarly, except that the transition between the Coulomb and Weertman regimes is continuous, which makes it easier to handle numerically (Brondex et al., 2017).

Although a growing number of information about the current state of the ice-sheet (e.g. surface velocities, surface elevation, surface elevation rates of change) are made available by the rapid development of satellite observations, several model parameters remain poorly constrained (e.g. bedrock elevation, ice viscosity, friction law coefficients). Gradient-based optimisation methods are routinely used to estimate uncertain model parameter fields and boundary conditions so that the initial ice-sheet geometry and velocity field are as close as possible to observations (e.g. Morlighem et al., 2010, 2013; Gillet-Chaulet et al.,

2012; Cornford et al., 2015). Although such methods enable to deduce the current basal shear stress field from observed surface velocities, the form of the friction law cannot be discriminated with a unique set of observations (Joughin et al., 2004; Gillet-Chaulet et al., 2016). Furthermore, when several uncertain fields are simultaneously inferred, several different initial states consistent with observations can potentially be constructed. Yet, Adhalgeirsdóttir et al. (2014) have shown that SLR projections on decadal timescales are sensitive to the model initial state which can account for an important source of uncertainty

in the model response.

    In the present study, we aim at assessing the relative sensitivity of centennial mass loss projections of the ASE to the chosen friction law and initialisation strategy. Our work being based on a schematic perturbation scenario, the results presented here should not be considered as actual projections of the future contribution of the ASE to SLR. To reach our goal, the first step consists in building three different model states of the ASE by inferring simultaneously the basal shear stress and the ice

viscosity, with various relative weights attributed to each one of these two fields during the inversion. Then, for each of these states, we follow the same procedure as in Brondex et al. (2017) to identify the distributions of the friction coefficients of three commonly used friction laws that lead to the same model initial states. Finally, we apply a synthetic perturbation of the basal melting rate to the different initial states and compare the dynamical responses obtained with the various friction laws. In Sect. 2, we give a precise description of the model used to conduct this study and describe the experimental setup. The results

obtained at each step of the experiments are presented in Sect. 3 and discussed in the last section.





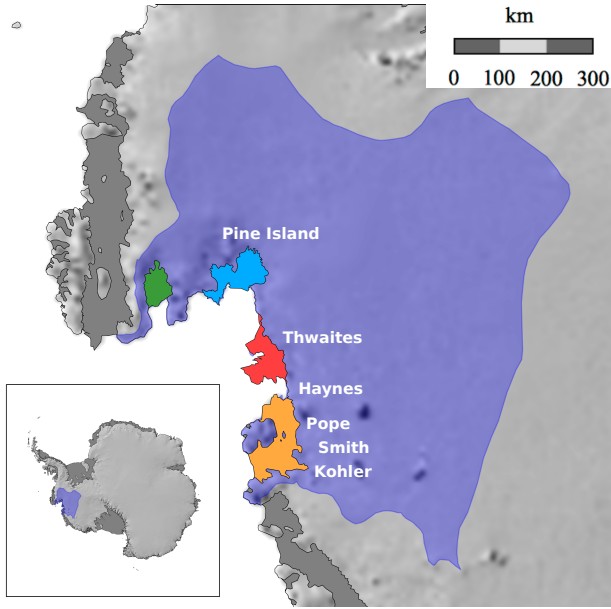

**Figure 1.** Amundsen Sea Embayment (ASE) and its ice shelves: Cosgrove (green), Pine Island (blue, fed by the Pine Island Glacier), Thwaites (red, fed by the Thwaites and Haynes ice streams), Crosson and Dotson (yellow, fed by the Pope, Smith and Kohler ice streams). The localisation of the ASE is reported on the map of Antarctica in the bottom left corner (purple).

## 2  Methods

### 2.1  Model description

The modeled domain is represented in Fig. 1. For the stress balance, we solve the two-dimensionnal shallow shelf approximation (SSA) equations (MacAyeal, 1989), for which the horizontal velocity field $\mathbf{u} = (u, v)$ is a solution of

$$
4\frac{\partial}{\partial x}\left(\bar{\eta}\frac{\partial u}{\partial x}\right) + 2\frac{\partial}{\partial x}\left(\bar{\eta}\frac{\partial v}{\partial y}\right) + \frac{\partial}{\partial y}\left(\bar{\eta}\left(\frac{\partial u}{\partial y} + \frac{\partial v}{\partial x}\right)\right)
$$
$$
- \tau_{b,x} = \rho_i g H \frac{\partial z_s}{\partial x}
$$
$$
4\frac{\partial}{\partial y}\left(\bar{\eta}\frac{\partial v}{\partial y}\right) + 2\frac{\partial}{\partial y}\left(\bar{\eta}\frac{\partial u}{\partial x}\right) + \frac{\partial}{\partial x}\left(\bar{\eta}\left(\frac{\partial u}{\partial y} + \frac{\partial v}{\partial x}\right)\right)
$$
$$
- \tau_{b,y} = \rho_i g H \frac{\partial z_s}{\partial y}, \tag{1}
$$

where $\rho_i$ is the ice density, $g$ the gravity norm and $H = z_s - z_b$ the ice thickness, with $z_s$ and $z_b$ the top and bottom surface elevations, respectively. The vertically integrated effective viscosity $\bar{\eta}$ reads:

$$
\bar{\eta} = \int_{z_b}^{z_s} \eta_0 D_e^{(1-n)/n} dz, \tag{2}
$$



where $D_e$ is the second invariant of the strain-rate tensor, $n$ is the Glen's law exponent and $\eta_0$ is the viscosity given by:

$$\eta_0 = \frac{1}{2} A^{-1/n}. \tag{3}$$

In Eq. (3), $A$ is the fluidity parameter. It is related to the temperature $T$ ($^\circ$ C) through an Arrhenius Law:

$$A = A_0 e^{(-Q/[R(273.15+T)])}, \tag{4}$$

where $A_0$ is the pre-exponential factor, $Q$ is an activation energy and $R$ is the gas constant.

For the basal shear stress $\boldsymbol{\tau}_b$ in Eq. (1), we consider three different friction laws:

$$\boldsymbol{\tau}_b + C_W |\mathbf{u}_b|^{m-1} \mathbf{u}_b = 0, \tag{5}$$

$$\boldsymbol{\tau}_b + C_B N |\mathbf{u}_b|^{m-1} \mathbf{u}_b = 0, \tag{6}$$

$$\boldsymbol{\tau}_b + \frac{C_S |\mathbf{u}_b|^{m-1} \mathbf{u}_b}{\left(1 + \left(\frac{C_S}{C_{max}N}\right)^{1/m} |\mathbf{u}_b|\right)^m} = 0, \tag{7}$$

where $C_W$, $C_B$ and $C_S$ are friction parameters. Equation (5) corresponds to the widely used Weertman law (Weertman, 1957), where $m$ is a positive exponent, often related to the creep exponent $n$ of the Glen's law as $m = 1/n$. Equations (6) and (7) correspond to the Budd and Schoof friction laws, respectively (Budd et al., 1979; Schoof, 2005). Note that, on the contrary to

the two other laws, the Schoof law induces an upper bound of the ratio $|\boldsymbol{\tau}_b|/N$ known as Iken's bound and equal to the value of the parameter $C_{max}$ (Iken, 1981; Schoof, 2005; Gagliardini et al., 2007). Although it is not its original purpose, the use of a Schoof law - which can be seen as the asymptotical equivalent of the Tsai law (Brondex et al., 2017) - is more and more justified by its ability to represent the deformation of till, as it induces a Coulomb friction regime at low $N$. In that case, the parameter $C_{max}$ ought to be seen as a Coulomb friction parameter related to the rheological properties of the till. Laboratory

measurements have shown that the values of $C_{max}$ should then range between $0.17$ and $0.84$ (Cuffey and Paterson, 2010).

Both the Budd and Schoof laws make an explicit use of the effective pressure $N$. Ideally, $N$ should be computed by a subglacial hydrology model but the few available models of that kind are usually relying on a multitude of poorly constrained parameters (Schoof, 2010; Hewitt et al., 2012; Werder et al., 2013; De Fleurian et al., 2014) and the attempts to couple these models to ice flow models are relatively recent (Hewitt, 2013; Bougamont et al., 2014; Bueler and van Pelt, 2015; Gagliardini

and Werder, 2018). Therefore, in the present study, $N$ is calculated assuming a perfect hydrological connection between the subglacial drainage system and the ocean, so that:

$$N = \begin{cases} \rho_i g H + \rho_w g z_b & \text{if } z_b < 0 \ , \\ \rho_i g H & \text{if } z_b \geq 0, \end{cases} \tag{8}$$





where $\rho_w$ is the water density. A systematic comparison of the dependence of $\boldsymbol{\tau}_b$ on $N$ and $\mathbf{u}_b$ implied by these various friction laws is available in Brondex et al. (2017). Note that, since floating ice does not experience any friction, $\boldsymbol{\tau}_b$ is set to zero wherever ice is afloat, independently of the chosen friction law.

The temporal evolution of ice thickness is governed by a two-dimensional mass transport equation:

$$5 \qquad \frac{\partial H}{\partial t} + \frac{\partial (uH)}{\partial x} + \frac{\partial (vH)}{\partial y} = a_s - a_b, \tag{9}$$

where $a_s$ is the meteoric accumulation rate applied to the top surface of the whole domain and $a_b$ the oceanic melt rate applied to the bottom surface of the ice shelves only. Ice is assumed to be in hydrostatic equilibrium and the bottom surface elevation can be deduced from the bedrock topography $b(x,y)$ by applying the no-penetration condition and the floating condition. Here we assume a constant sea level $z_{sl} = 0$, so that:

$$10 \qquad \begin{cases} z_b(x,y,t) = b(x,y) & \text{for grounded ice,} \\ z_b(x,y,t) = -H \frac{\rho_i}{\rho_w} > b(x,y) & \text{for floating ice.} \end{cases} \tag{10}$$

The GL being the limit beyond which grounded ice starts floating, its position $(x_G, y_G)$ can directly be deduced from Eq. (10) by solving:

$$H(x_G, y_G) + b(x_G, y_G) \frac{\rho_w}{\rho_i} = 0. \tag{11}$$

Therefore, the GL can be located at any point of the domain and its position is free to evolve over time as a result of evolving 15 geometry.

The positions of the boundaries of the model domain are set based on the observations made by the satellite ICESat for the IMBIE2 effort (Zwally et al., 2012). In the following, $\mathbf{n} = (n_x, n_y)$ is the normal unit vector to the considered boundary. At the ice divides, there is no ice flux entering the domain so that the following Dirichlet condition applies:

$$(\mathbf{u}.\mathbf{n})|_{id} = 0, \tag{12}$$

20 and is completed by a free slip condition in the tangential direction. At the ice shelves and glaciers fronts, the following Neumann condition applies:

$$4\bar{\eta} \left.\frac{\partial u}{\partial x}\right|_f n_x + 2\bar{\eta} \left.\frac{\partial v}{\partial y}\right|_f n_x + \bar{\eta} \left.\left(\frac{\partial u}{\partial y} + \frac{\partial v}{\partial x}\right)\right|_f n_y$$
$$= \frac{g}{2}(\rho_i H|_f^2 - \rho_w H_{sub}|_f^2)n_x$$
$$2\bar{\eta} \left.\frac{\partial u}{\partial x}\right|_f n_y + 4\bar{\eta} \left.\frac{\partial v}{\partial y}\right|_f n_y + \bar{\eta} \left.\left(\frac{\partial u}{\partial y} + \frac{\partial v}{\partial x}\right)\right|_f n_x$$
$$= \frac{g}{2}(\rho_i H|_f^2 - \rho_w H_{sub}|_f^2)n_y, \tag{13}$$

where $H_{sub}|_f$ is the submerged height at the considered front, which is null in the very few places where glacier fronts are grounded and relates to $H|_f$ through the floating condition at floating fronts.





Similarly to Pollard and DeConto (2012), the oceanic melt rate $a_b$ (m a$^{-1}$), prescribed on the bottom surface of ice shelves only, is parameterised as follows:

$$a_b = \frac{8K_T \rho_w c_w}{\rho_i L_f} |T_0 - T_f|(T_0 - T_f), \tag{14}$$

where $T_0$ and $T_f$ are, respectively, the ocean water temperature and freezing point at the considered depth, $K_T$ is a transfer

coefficient for sub-ice oceanic melting, $c_w$ the specific heat of ocean water and $L_f$ the latent heat of fusion of ice. Following Pollard and DeConto (2012), the temperature difference $T_0 - T_f$ is given by: $T_0 - T_f = 0.5\,^{\circ}$ C for $0 > z > -170$ m, $3.5\,^{\circ}$ C for $z < -680$ m and linearly interpolated for $z$ between $-680$ and $-170$ m. Values of parameters prescribed in this study are presented in Table 1.

For the surface mass balance $a_s$, we use outputs of simulations performed with the MAR model and averaged over the

1979-2015 period (Agosta, personal communication). The bedrock topography is taken from Bedmap2 (Fretwell et al., 2013), except that we include two pinning points in contact with the bottom surface of Thwaites ice shelf using the bathymetry of Millan et al. (2017). The mechanical role of these two pinning point has indeed been shown to be of first importance because of the buttressing effect that they exert on the upstream ice stream (Fürst et al., 2016).

All the equations presented above are solved using the open source finite element code Elmer/Ice (Gagliardini et al., 2013).

The mesh is generated using the anisotropic mesh-adaptation technique described in Gillet-Chaulet et al. (2012) so that the distance between two nodes at the GL is of the order of $\sim 50$ m. Finally, we end up with a mesh made of 274678 nodes covering the whole model domain, the total surface of which is about $4.5 \times 10^5$ km$^2$. The same mesh is used for all the successive steps of all the experiments presented in the following. In addition, we take advantage of the fact that the GL position is determined at subelement precision through Eq. (11) to make use of the subelement parameterisation SEP3 as proposed by Seroussi et al.

(2014). In our case, the number of quadrature points is raised to 20 in the elements containing the GL.

## 2.2 Experimental setup

In the present study, we aim at comparing the results in terms of GL dynamics and volume losses produced by the following friction laws: (1) a linear Weertman law which corresponds to Eq. (5) with $m = 1$, (2) a non-linear Weertman law given by Eq. (5) with $m = 1/3$, (3) a non-linear Budd law given by Eq. (6) with $m = 1/3$, (4) a non-linear Schoof law given by Eq. (7)

with $m = 1/3$ and $C_{max} = 0.4$, and (5) a non-linear Schoof law given by Eq. (7) with $m = 1/3$ and $C_{max} = 0.6$. To reach this goal, we start with an initialisation step in which three model states - hereinafter referred to as inferred states - with different initial basal shear stress and viscosity fields are constructed from available observations, before being submitted to a 15-year relaxation period. Then, following the same procedure as the one described in Brondex et al. (2017) for each of these three inferred states, we identify the distributions of the friction coefficients of the aforementioned friction laws so that the basal

shear stress field computed with these laws is the same as the one obtained at the end of the 15-year relaxation period. After this step, we are left with 13 new model states, hereinafter referred to as initial states, corresponding to four friction laws - the linear and non-linear Weertman laws and the non-linear Schoof laws with $C_{max} = 0.4$ and $C_{max} = 0.6$ - applied to the three inferred states as well as the Budd law applied to one of the inferred state only. These 13 initial states constitute the starting



**Table 1.** List of parameter values used in this study.

| Parameters | Values | Units |
|---|---|---|
| $\rho_i$ | 910 | kg m$^{-3}$ |
| $\rho_w$ | 1028 | kg m$^{-3}$ |
| $A_0(T < -10\,^\circ\text{C})$ | $2.847 \times 10^{-13}$ | Pa$^{-3}$ s$^{-1}$ |
| $A_0(T > -10\,^\circ\text{C})$ | $2.360 \times 10^{-2}$ | Pa$^{-3}$ s$^{-1}$ |
| $c_w$ | 4.218 | kJ kg$^{-1}$ K$^{-1}$ |
| $g$ | 9.81 | m s$^{-2}$ |
| $K_T$ | 15.77 | m a$^{-1}$ K$^{-1}$ |
| $L_f$ | 335 | kJ kg$^{-1}$ |
| $n$ | 3 | |
| $Q(T < -10\,^\circ\text{C})$ | 60 | kJ mol$^{-1}$ |
| $Q(T > -10\,^\circ\text{C})$ | 115 | kJ mol$^{-1}$ |
| $R$ | 8.314 | J mol$^{-1}$ K$^{-1}$ |

points of a set of two $100 + 5$ year pronostic simulations, i.e. an unperturbed control run and a run for which the oceanic melt rate is perturbed. Figure 2 summarises all the consecutive steps of the experimental setup which are described in details below.

### 2.2.1 Initialisation

A number of observations are used in order to initialise the model. The initial ice thickness is taken from Bedmap2 (Fretwell et al., 2013). We use the ice temperature map of Antarctica established by Van Liefferinge and Pattyn (2013) in order to calculate a reference viscosity field $\eta_{0,ref}$ from Eqs. (3) and (4). In addition, we use the observed surface velocities of Rignot et al. (2011) in order to formulate an inverse problem. These velocities are given on each nodes of a regular grid with a $450$ m resolution, except in some regions, sometimes very wide, over which the information is missing. The estimated error on the norm of observed velocities ranges, depending on regions, between $1$ m a$^{-1}$ and $17$ m a$^{-1}$. These observations have been collected over periods spanning several years which could potentially induce inconsistencies in regions where the flow features are evolving rapidly.

The flow solution **u** of the SSA equations (1) depends on both the basal shear stress field and the viscosity field. For simplicity, the initial basal shear stress field is computed with a linear Weertman law of which we infer the friction coefficient distribution $\hat{C}_W$. Although $\eta_{0,ref}$ constitutes a first approximation of ice viscosity, there are significant uncertainties regarding the temperature map of Van Liefferinge and Pattyn (2013) as well as the parameters linking the ice fluidity to its temperature. In addition, ice viscosity does not depend only on temperature but also on several other parameters including damage, anisotropy, water content, density, grain size and impurities content (Cuffey and Paterson, 2010). Therefore, the question is wether the



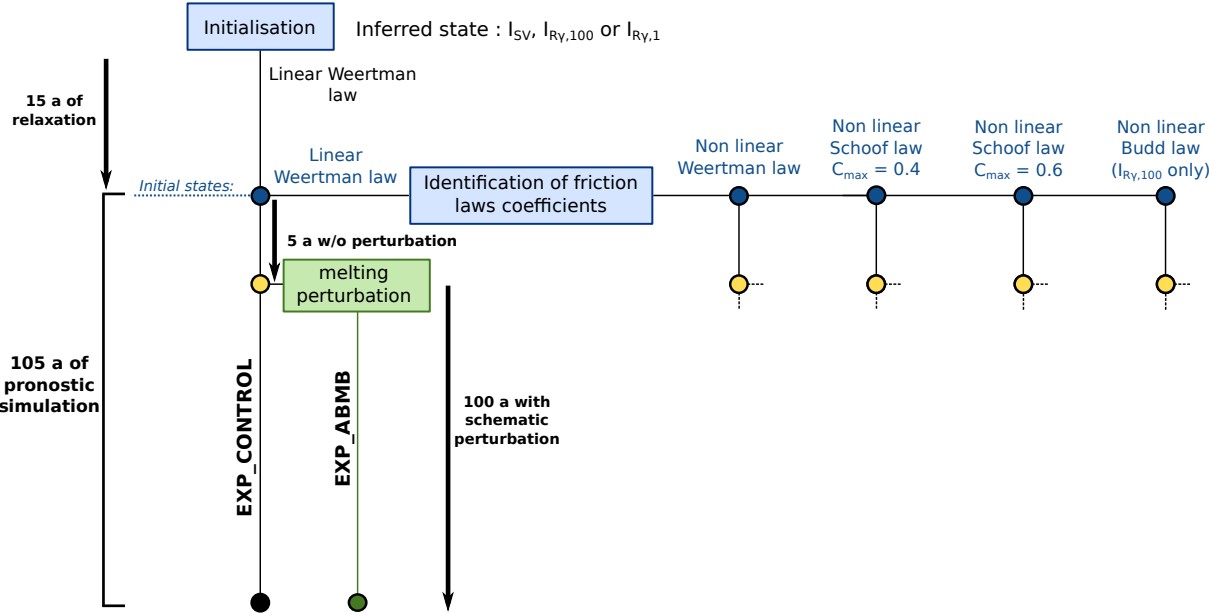

**Figure 2.** Flowchart summarising the consecutive steps of the present study for a given inferred state. The simulations EXP_CONTROL and EXP_ABMB are run for each of the five friction laws. This same procedure is repeated for each of the three inferred states $I_{SV}$, $I_{R_{\gamma,100}}$ and $I_{R_{\gamma,1}}$, except that the Budd friciton law is applied to the inferred state $I_{R_{\gamma,100}}$ only.

local discrepancies between observed and modelled velocities at any given point of the domain should be attributed to an inappropriate basal shear stress field or to errors on the reference viscosity field $\eta_{0,ref}$. Indeed, several model states consistent with observations can be constructed, for which the fit between modelled and observed velocities is obtained by adjusting rather the basal shear stress or rather the viscosity. In order to assess the sensitivity of the results to the initialisation strategy,

5   we construct three inferred states - denoted $I_{SV}$, $I_{R_{\gamma,100}}$ and $I_{R_{\gamma,1}}$ - by means of the control method, building on the approach described in Fürst et al. (2015).

The total cost function $J_{tot}$ to minimise includes two cost functions and two Tikhonov regularisation terms:

$$J_{tot} = J_v + \lambda_{div} J_{div} + \lambda_{reg,\alpha} J_{reg,\alpha} + \lambda_{reg,\gamma} J_{reg,\gamma}. \tag{15}$$

The misfit between modelled ($\mathbf{u}$) and observed ($\mathbf{u}_{obs}$) velocities is comprised in the first cost term $J_v$. To avoid errors due to

10   interpolation of observed velocities at the model mesh nodes, $J_v$ is a discrete sum evaluated directly at the observation points. It reads:

$$J_v = \frac{1}{2} \sum_{i=1}^{N_{obs}} (|\mathbf{u}_{i,obs} - \mathbf{u}_i|)^2, \tag{16}$$

where $N_{obs}$ is the total number of available observations, $\mathbf{u}_{i,obs}$ the velocity observed at point $i$ and $\mathbf{u}_i$ the modelled velocity which is interpolated at the observation point $i$. The second cost function $J_{div}$ is intended to penalise the large gaps between





ice flux divergence and mass balance, leading to inferred states closer to steady states. It reads:

$$J_{div} = \frac{1}{2} \int_{\Gamma} \left( \frac{\partial(uH)}{\partial x} + \frac{\partial(vH)}{\partial y} - (a_s - a_b) \right)^2 d\Gamma, \tag{17}$$

where $\Gamma$ is the model domain. During the inversion, we actually optimise the variables $\alpha$ and $\gamma$ which are related to, respectively, the linear Weertman law coefficient and the viscosity as follows:

$$\hat{C}_W = 10^{\alpha}, \tag{18}$$

and

$$\eta_0 = \frac{\gamma^2}{R_\gamma} \eta_{0,ref}, \tag{19}$$

where $R_\gamma$ is a constant. The variable changes (18) and (19) prevent non-physical negative values of basal shear stress and viscosity, respectively. In addition, the variable change (19) enables to tune the relative weight that will be put on basal shear stress and viscosity during inversion. Indeed, the minimisation of the cost functions relies on the calculation of their gradients with respect to the variables to optimise (MacAyeal, 1993). As a consequence, the highest the value attributed to $R_\gamma$ the lowest the gradients of $J_v$ and $J_{div}$ calculated with respect to $\gamma$ relatively to the ones calculated with respect to $\alpha$. In such a case, the distribution of $\alpha$ - and thus of the basal shear stress - will be more affected (on the grounded part only as $\boldsymbol{\tau}_b$ is forced to zero wherever ice is floating) than for lower values of $R_\gamma$. The two regularisation functions $J_{reg,\alpha}$ and $J_{reg,\gamma}$ in turn penalise first spatial derivatives of, respectively, $\alpha$ and $\gamma$. They are meant to avoid overfitting the velocity observations and thus, to improve the conditioning of the problem.

Among the three inferred states considered in this study, the two states $I_{R_{\gamma,100}}$ and $I_{R_{\gamma,1}}$ are constructed by optimising both $\alpha$ and $\gamma$ with, respectively, $R_\gamma = 100$ and $R_\gamma = 1$ in Eq. (19). Hence, more weight is put on basal shear stress to the detriment of viscosity for $I_{R_{\gamma,100}}$ than for $I_{R_{\gamma,1}}$. In contrast, the inferred state $I_{SV}$ is obtained by optimising $\alpha$ only, while the viscosity is forced to $\eta_0 = \eta_{0,ref}$. The gradients of $J_{tot}$ are derived following the adjoint method (MacAyeal, 1993), while the minisation itself is done using the quasi-Newton routine M1QN3 (Gilbert and Lemaréchal, 1989). The minimisation method being an iterative method, we need to provide initial guesses for the distributions of the variables to optimise. Following Fürst et al. (2015), the initial guess for $\alpha$ is found by considering that, as a first approximation, the driving stress is exactly compensated by the basal shear stress at any given point of the ice/bed interface. Regarding the initial field of viscosity, it is simply set to $\eta_{0,ref}$ for both $I_{R_{\gamma,100}}$ and $I_{R_{\gamma,1}}$.

For each of the three inferred states, we follow a L-curve approach as described in Fürst et al. (2015) to retrieve the optimal values of $\lambda_{div}$, $\lambda_{reg,\alpha}$ and $\lambda_{reg,\gamma}$ (only for $I_{R_{\gamma,100}}$ and $I_{R_{\gamma,1}}$) leading to inferred states showing good compromise between smooth distributions of $\alpha$ and $\gamma$, low differences between ice fluxes and mass balance and good fit between observed and modelled velocities. We find, respectively, $(5.1 \times 10^{-5}, 7.1 \times 10^5)$ for $I_{SV}$, $(5.9 \times 10^{-5}, 2.5 \times 10^6, 1.4 \times 10^6)$ for $I_{R_{\gamma,100}}$, and $(1.8 \times 10^{-5}, 5.4 \times 10^6, 4.7 \times 10^5)$ for $I_{R_{\gamma,1}}$.

Although strong discrepancies between mass balance and ice flux are penalised through $J_{div}$ in Eq. (15), the three inferred states are not exactly steady states. Seroussi et al. (2011) and Gillet-Chaulet et al. (2012) have reported the occurence of non-





physical ice thickness rates of change in the first years following inversion. These ice flux anomalies are due to uncertainties on the model initial conditions, in particular regarding the prescribed topography and the values attributed to the various parameters, and are usually dissipated within a few years. For this reason, each of the three inferred states is submitted to a relaxation period, the duration of which is arbitrarily fixed to 15 years. These relaxations are performed with the direct model

as described in Sect. 2.1, the basal shear stress field being computed through the linear Weertman law (see Fig. 2).

### 2.2.2   Identification of friction laws coefficients

For each of the three inferred states, the 15-year relaxation period leads to a new state characterised by a new velocity field, which we denote $\hat{\mathbf{u}}_b$, as well as a new basal shear stress field, denoted $\hat{\boldsymbol{\tau}}_b$. These two fields relate to each other through the linear Weertman law, i.e. Eq (5) with the inferred friction coefficient $\hat{C}_W$ (which depends on the considered inferred state) and

$m = 1$. As thoroughly described in Brondex et al. (2017), these two fields can be used to analytically identify the distribution of the friction coefficient of another law which would lead to the same reference basal shear stress field $\hat{\boldsymbol{\tau}}_b$. Thus, the distributions of the friction coefficient of the non-linear Weertman law, the non-linear Budd law and the Schoof law can be identified by solving, respectively:

$$C_{W,nl} = \hat{C}_W |\hat{\mathbf{u}}_b|^{1-m},\tag{20}$$

$$C_B = \hat{C}_W \frac{|\hat{\mathbf{u}}_b|^{1-m}}{\hat{N}},\tag{21}$$

and,

$$C_S = \frac{|\hat{\boldsymbol{\tau}}_b|}{|\hat{\mathbf{u}}_b|^m \left(1 - \left(\frac{|\hat{\boldsymbol{\tau}}_b|}{C_{max}\hat{N}}\right)^{1/m}\right)^m}.\tag{22}$$

For the Budd and Schoof laws, the effective pressure field $\hat{N}$ used for the identification is calculated through Eq. (8) from

the geometry obtained at the end of the relaxation period. Note that the fields $\hat{C}_W$, $\hat{\mathbf{u}}_b$, $\hat{\boldsymbol{\tau}}_b$ and $\hat{N}$ being dependent of the inferred state, these identifications are repeated for each one of the three inferred states, except for the Budd law for which the identification has been done only for the case $I_{R_{\gamma,100}}$ (see Fig. 2). Equations (20) and (21) enable to identify the friction coefficients of, respectively, the non-linear Weertman and Budd laws at every grounded nodes covered with ice. At floating nodes or in the very few places of the domain which are ice free when identification is performed, these coefficients are

arbitrarily fixed to, respectively, $C_{W,nl} = 10^{-6}$ MPa m$^{-1/3}$ a$^{1/3}$ and $C_B = 10^{-6}$ m$^{-1/3}$ a$^{1/3}$. This choice does not appear to be critical as all the following prognostic experiments lead to a GL retreat.

In contrast, because the Schoof law was contructed so that Iken's bound is satisfied (Schoof, 2005), Eq. (22) has a solution only in places where $|\hat{\boldsymbol{\tau}}_b|/\hat{N} < C_{max}$. Figure 3 shows, for each of the three inferred states, the percentage of grounded nodes where the value of $C_S$ cannot be identified directly from Eq. (22), depending on the value attributed to the parameter $C_{max}$.

Although we do not expect the parameter $C_{max}$ to be uniform all over the bed, there is currently no way to constrain its





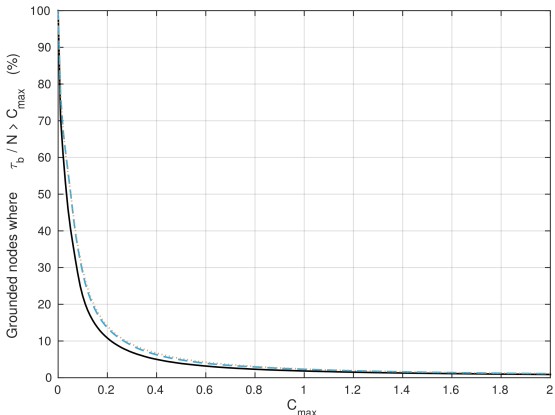

**Figure 3.** Percentage of grounded nodes where Iken's bound is not satisfied as a function of the value attributed to the parameter $C_{max}$, for the inferred states $I_{SV}$ (black solid line), $I_{R\gamma,100}$ (blue dashed line) and $I_{R\gamma,1}$ (brown dotted line).

spatial distribution. As a consequence, two different uniform values are considered: $C_{max} = 0.4$ and $C_{max} = 0.6$. Note that the identification of $C_S$ has then to be done for each of the two values of $C_{max}$. Because the sensitivity of basal shear stress to $C_S$ is very small in places close to flotation, the identification of $C_S$ from Eq. 22 is only done at nodes where $|\hat{\tau}_b|/\hat{N} \leq 0.8C_{max}$. The values of $C_S$ at nodes where $|\hat{\tau}_b|/\hat{N} > 0.8C_{max}$ are linearly interpolated from the closest neighboring nodes. In addition,

we arbitrarily set $C_S = 10^{-3}$ MPa m$^{-1/3}$ a$^{1/3}$ in places which are ice free or where ice is afloat when the identification is performed. Finally, we end up with six different distributions of $C_S$ corresponding to the two values of $C_{max}$ applied to the three inferred states $I_{SV}$, $I_{R\gamma,100}$ and $I_{R\gamma,1}$. For these six distributions, the percentage of grounded nodes at which $C_S$ has to be interpolated is comprised between $\sim 4\,\%$ (for the initial state $I_{SV}$ associated to the parameter $C_{max} = 0.4$) and $\sim 8\,\%$ (for the initial state $I_{R\gamma,1}$ associated to the parameter $C_{max} = 0.6$). As expected, the nodes at which an interpolation is required are

the closest to flotation and, therefore, are mostly located right upstream the GL.

As stated previously, the identification steps lead to 13 different initial states which are used as the starting points of the prognostic simulations (blue dots in Fig. 2).

### 2.2.3 Prognostic simulations

The 13 initial states are submitted to a set of two 105-year prognostic runs: (i) a control run (EXP_CONTROL) and (ii) a

basal melt anomaly run (EXP_ABMB). The EXP_CONTROL run is an unforced forward experiment aiming at characterizing model drift which depends both on the initial state and the friction law. Therefore, all model parameters and forcing in EXP_CONTROL are the same as those used for initialisation and presented in Sect. 2.1.

The EXP_ABMB run consists in applying a synthetic anomaly of basal melting rate under floating ice, all other model parameters and forcing being kept the same as in EXP_CONTROL: after a first 5-year period free of any perturbation, a

uniform basal melting rate anomaly $abmb$ of $13.2$ m a$^{-1}$ is progressively added to the initial basal melting rate $a_b$, given by



Eq. (14), over the following $40$ years of simulation. Thus, the basal melting rate $a_{b,ABMB}$ for this run is given by:

$$\begin{cases} a_{b,ABMB} = a_b & \text{if } 0 \leq t < 5\,\text{a} \\ a_{b,ABMB} = a_b + abmb\frac{\text{floor}(t)-5}{40} & \text{if } 5 \leq t < 45\,\text{a} \\ a_{b,ABMB} = a_b + abmb & \text{if } t \geq 45\,\text{a}. \end{cases} \qquad (23)$$

The basal melting rate anomaly $abmb$ corresponds to the one prescibed for the ASE within the InitMIP-Antarctica framework (Nowicki et al., 2016).

## 3 Results

### 3.1 Initialisation

The three inferred states $I_{SV}$, $I_{R_{\gamma,100}}$ and $I_{R_{\gamma,1}}$ are compared in Fig. 4. The absolute difference between modelled $\mathbf{u}$ and observed $\mathbf{u}_{obs}$ velocities turns out to get smaller when both basal shear stress and viscosity are inferred (Fig. 4a-c). This is particularly true for the ice shelves which do not feel any basal shear stress: although the basal shear stress right upstream the GL do influence velocities within the downstream ice shelf, the most efficient way to get a better match between modelled and observed velocities in floating areas is through a local adjusement of viscosity. Thus, the inferred state $I_{SV}$ shows important errors on the Pine Island, Thwaites and, to a lesser extent, Crosson ice shelves (Fig. 4a). The root mean squared (RMS) errors obtained for the inferred states $I_{SV}$, $I_{R_{\gamma,100}}$ and $I_{R_{\gamma,1}}$ are, respectively, $85.5\,\text{m a}^{-1}$, $52.4\,\text{m a}^{-1}$ and $37.5\,\text{m a}^{-1}$.

The ratio between the norm of the basal shear stress and the norm of the driving stress is shown in Fig. 4d-f. At the end of the inversions, there are several regions where the local driving stress is not entirely compensated by the local basal shear stress (blue regions). As expected, these regions correspond to regions of rapid flow; in particular, Pine Island Ice Stream can be easily distinguished. The regions of low $\boldsymbol{\tau}_b$ gets narrower as more weight is put on viscosity during inversion. Indeed, when the viscosity is inferred, another way for the inversion algorithmn to increase the modelled velocities in areas where they would be too low otherwise is to soften the ice locally: this is the case, for example, in the higher part of PIG, in particular for the inferred state $I_{R_{\gamma,1}}$ for which the inversion algorithm induces a local reduction of viscosity rather than of the basal stress to increase the modelled velocities (panels f and i of Fig. 4). It is also this same mechanism which is behind the low viscosity bands which can easily be distinguished in Fig. 4h (note that they are also present in Fig. 4i but less visible due to the wider color scale). These soft bands correspond to shear margins where shear stresses are high and, therefore, ice is strongly damaged. The correlation between highly fractured areas and locally reduced viscosity has already been showed in several previous studies (Vieli et al., 2006, 2007; Khazendar et al., 2007; Borstad et al., 2012, 2013). On the contrary, the inversion algorithm can render the ice locally stiffer in order to decrease the modelled velocities in areas where they are too high compared to observations, which is typically the case for ice shelves.

Because the model states obtained after the initialisation step are not steady states, they drift toward new states during the following 15-year relaxation period. Overall, the three inferred states show similar evolution patterns with relatively moderate ice thickness changes, except for the Pine Island and Thwaites ice shelves where ice thickness increases of about $100\,\text{m}$ over the





15 years. In contrast, ice streams get slightly thinner, loosing a few tens of m of ice in some places. At the end of the relaxation period, the thickness rates of change have decreased to physically acceptable values for all the inferred states, with, at most, $30\,\mathrm{m\,a^{-1}}$ of absolute thickness rate of change, concentrated within very localised areas; elsewhere, ice thickness is nearly at equilibrium with, at most, a few $\mathrm{m\,a^{-1}}$ of absolute thickness rate of change. Although the modelled velocity structure remains

similar to observations, the RMS errors between observed velocities and velocities computed at the end of the relaxation period have risen to $124.8\,\mathrm{m\,a^{-1}}$, $119.4\,\mathrm{m\,a^{-1}}$ and $96.4\,\mathrm{m\,a^{-1}}$ for, respectively, $I_{SV}$, $I_{R_{\gamma,100}}$ and $I_{R_{\gamma,1}}$.

### 3.2   Identification of friction laws coefficients

For each of the three inferred states, the basal shear stress fields computed with the non-linear Weertman and Budd laws using the friction coefficient distributions deduced through, respectively, Eqs. (20) and (21) are identical to the basal shear stress

field $\hat{\boldsymbol{\tau}}_b$ used for the identifications. In contrast, because the value of $C_S$ cannot be exactly identified through Eq. (22) at every grounded nodes but need to be interpolated in some places, $\hat{\boldsymbol{\tau}}_b$ is not perfectly reproduced with the Schoof law. Figure 5a-c show an enlargement of regions within which the relative differences between the basal shear stress field calculated with the non-linear Schoof law (7) associated to $C_{max} = 0.4$ and to the $C_S$ distributions identified through Eq. (22) or interpolated, which is denoted $\boldsymbol{\tau}_{bS04}$, and the reference basal shear stress field $\hat{\boldsymbol{\tau}}_b$ are the highest. As expected, the highest relative differences

are obtained close to the ice shelves (shown in green in Fig. 5a-c) where the nodes are close to flotation and $C_S$ needs to be interpolated. Elsewhere, the fields of $\boldsymbol{\tau}_b$ produced by the two friction laws are numerically identical. The relative difference between $\boldsymbol{\tau}_{bS04}$ and $\hat{\boldsymbol{\tau}}_b$ at nodes where $C_S$ is interpolated seldom exceeds $10\,\%$; higher differences (up to $100\,\%$) occur very locally at some of the last grounded nodes. The results obtained with the parameter $C_{max} = 0.6$ (not shown) are similar except that the nodes where the interpolation of $C_S$ is required are fewer and, therefore, the regions whithin which the relative

difference between $\boldsymbol{\tau}_{bS06}$ and $\hat{\boldsymbol{\tau}}_b$ is not null are less extended. Note that in places where the relative difference is not null, the Schoof law systematically produces the weakest basal shear stress.

Although moderate and very localised, the differences between the basal shear stress fields computed with the linear Weertman law and the Schoof law induce significant differences on the recomputed velocity fields (Fig. 5d-f). These differences propagate from the regions showing the highest differences in terms of $\boldsymbol{\tau}_b$ to the floating areas located right downstream (par-

ticularly distinguishable on Thwaites Ice Shelf). Indeed, the decrease of the total basal shear stress due to the interpolation of $C_S$ close to the GL needs to be compensated elsewhere so that the global stress balance keeps being satisfied. Because the floating areas do not support any basal shear stress, the perturbation is transmitted to the front of ice shelves, unless it can be compensated by an increase of the buttressing effect through a contact on a pinning point or an increase of lateral stresses at shear margins in the case of confined ice shelves. This latter mechanism is well-illustrated in Fig. 5d-f, which shows an

important increase of velocities at the shear margins of PIG when the linear Weertman law is replaced by the Schoof law. Furthermore, it turns out that the relative differences on the velocity field are more pronounced when more weight is put on basal shear stress during the initialisations. Indeed, when the inversion focus on the basal shear stress field only (inferred state $I_{SV}$), the ice flow is more sensitive to any perturbation of this field. Once again, the results obtained with the parameter $C_{max} = 0.6$ (not shown) are very similar with slightly lower relative differences between the velocity fields calculated with the two laws.



Finally, the identification (22) being performed at mesh nodes, the gaussian integration on elements induces small numerical errors on the recomputed velocity field. These numerical errors are concentrated in coarse mesh areas and correspond to the discrepancies of, at most, a few tens of percents observed further inland, in places where the differences in terms of $\boldsymbol{\tau}_b$ are null (Fig. 5d-f). Similar numerical errors are observed within the same areas when the velocity fields produced with the non-linear

Weertman and Budd laws after the identification step are compared to $\hat{\mathbf{u}}_b$.

### 3.3 Prognostic simulations

The evolution of the grounded ice area during the control run for the three inferred states and the various friction laws is shown in Fig. 6. A decrease (resp. an increase) of the grounded ice area reflects a retreat (resp. an advance) of the GL. After relaxation, the grounded ice area at $t = 0$ a differs depending on the considered inferred state: it ranges between 433400 $\mathrm{km}^2$ for $I_{R_{\gamma,1}}$

and 434000 $\mathrm{km}^2$ for $I_{SV}$.

Independently of the chosen inferred state or friction law, the natural (i.e. unperturbed) evolution of the domain is toward a reduction of its grounded ice area: the Schoof law with $C_{max} = 0.4$ systematically gives the more pronounced GL retreat with a reduction of grounded ice area over the 105 a of the control run ranging between 5000 $\mathrm{km}^2$ for $I_{R_{\gamma,1}}$ and 7700 $\mathrm{km}^2$ for $I_{SV}$. In contrast, the reduction of the grounded ice area obtained with the non-linear Weertman law ranges between 2400 $\mathrm{km}^2$ for

$I_{R_{\gamma,100}}$ and 4300 $\mathrm{km}^2$ for $I_{SV}$. Note that the evolutions of the grounded ice area obtained with the two Weertman laws are very close to one another all over the 105 a of the control run while the two Schoof laws produce noticeably different evolutions, with the case $C_{max} = 0.6$ showing less GL retreat than the case $C_{max} = 0.4$. This is very likely due to the differences between $\hat{\mathbf{u}}_b$ and the velocity field recomputed with the Schoof law after initialisation, which are lower for $C_{max} = 0.6$ than for $C_{max} = 0.4$ as reported in Sect. 3.2. For the inferred state $I_{R_{\gamma,100}}$, the Budd law produces an intermediate result with a reduction of the

grounded ice area of 3000 $\mathrm{km}^2$.

The results of the perturbation experiments EXP_ABMB are shown relative to the control simulation EXP_CONTROL (Fig. 7). We consider both the relative reductions of grounded ice area as well as the relative losses of volume above flotation (VAF), given in mm of sea level equivalent (SLE). For the inferred state $I_{SV}$, the grounded ice areas produced by the two Weertman laws at the end of EXP_ABMB are about 2000 $\mathrm{km}^2$ larger than the ones produced by the two Schoof laws. The

difference between the linear and non-linear Weertman laws or between the two Schoof laws ($C_{max} = 0.4$ and $C_{max} = 0.6$) is about ten times smaller ($\sim 200 \mathrm{km}^2$). However, the differences in terms of VAF change are more significant with, respectively, 6 mm SLE of difference between the two Weertman laws and 3 mm SLE of difference between the two Schoof laws. For comparison, the linear Weertman law and the Schoof law with $C_{max} = 0.4$ show 24 mm SLE of difference in terms of VAF loss at the end of EXP_ABMB. When the viscosity is adjusted during inversion (cases $I_{R_{\gamma,100}}$ and $I_{R_{\gamma,1}}$), the GL shows less

retreat and the relative VAF losses are less important (Fig. 7b-c and Fig. 7e-f). In addition, the relative evolutions of VAF obtained with the Weertman laws get closer to the ones obtained with the Schoof laws, except for the experiment associated to the inferred state $I_{R_{\gamma,1}}$ and the linear Weertman law (Fig. 7e-f). Note that the Schoof laws produce more GL retreat and more SLR contribution than the Weertman laws for the inferred states $I_{SV}$ and $I_{R_{\gamma,100}}$. On the contrary, for the inferred state $I_{R_{\gamma,1}}$, the grounded ice areas seem to stabilise around $t = 60$ a with the Schoof laws while they keep decreasing for a few



more years with the Weertman laws, so that the grounded ice areas obtained with the two latter at the end of EXP_ABMB are slightly less than the ones obtained with the two former. Despite a very similar decrease in terms of grounded ice areas as the ones obtained with the two Schoof laws, the linear Weertman law predicts 8 mm SLE less VAF loss (Fig. 7f). For the inferred state $I_{R_{\gamma,100}}$, the Budd friction law shows a much stronger response to the basal melting rate perturbation than the four other

laws: at the end of EXP_ABMB, it results in a grounded ice area reduced by 13000 km$^2$ relative to EXP_CONTROL, while the grounded ice area reduction ranges between 6300 km$^2$ and 8900 km$^2$ for the four other experiments. Likewise, the relative VAF loss obtained with the Budd law at the end of EXP_ABMB is of 33 mm SLE, whereas it ranges between 6 mm SLE and 15 mm SLE for the other experiments. This dramatic response of the GL dynamics to the perturbation when the Budd law is used is in line with the results reported in Brondex et al. (2017).

The GL positions obtained with the various initial states at the beginning and at the end of experiment EXP_ABMB are represented in Fig. 8. The GL produced with the two Schoof laws are almost systematically more retreated than the ones produced with the two Weertman laws. On the other hand, the GL final positions obtained with the Schoof law associated to $C_{max} = 0.6$ (brown solid lines) are generally very close to the ones obtained with the Schoof law associated to $C_{max} = 0.4$ (green solid lines). Likewise, the GL final positions obtained with the linear Weertman law (cyan solid lines) are often very

close to the ones obtained with the non-linear Weertman law (magenta solid lines), except for the inferred state $I_{SV}$ where the former is sometimes more advanced and sometimes more retreated than the latter. Note that this does not contradict the previously mentioned result regarding the similar evolution of grounded ice areas obtained with the two Weertman laws for $I_{SV}$, as evolutions represented in Fig. 7 are relative to EXP_CONTROL while the GL positions reported in Fig. 8 are absolute. The spatial distribution of the GL retreat is primary controlled by the bed topography, as already reported by Seroussi et al.

(2017). As a consequence, the Schoof and Weertman laws all give the same retreat patterns: the most pronounced retreats are observed in regions of gentle slope (particularly visible upstream of the Thwaites Ice Shelf) while the GL tends to wrap around prominences. However, for $I_{R_{\gamma,100}}$, the pattern of GL retreat obtained with the Budd law is significantly different than the ones obtained with the four other laws. Indeed, the final GL position obtained with the Budd law is the most advanced in the Thwaites region, where the other laws (especially the Schoof laws) induce important retreats. On the contrary, the Budd law

induces retreats of several tens of km in the regions of Cosgrove and Dotson Ice Shelves, where the four other laws give very similar final GL positions with limited retreat (especially in the Cosgrove Ice Shelf region).

The initial and final ice-sheet profiles obtained with the Schoof and Budd laws are represented in Fig. 9, along four selected flowlines (reported in white in Fig. 8b). By definition, the ice thickness comprised between $z_s$ and the flotation altitude $z_f$, given by $z_f = (1 - \rho_w/\rho_i)b$, constitutes the thickness above flotation. Because the GL position is deduced from Eq. (11), it is

located at the exact vertical of the point where $z_s = z_f$. Looking at the initial profiles, it turns out that the initial ice thickness above flotation increases rapidly upstream of the GL for the PIG and Thwaites flowlines. On the contrary, it increases very progressively as going further upstream within the grounded part for the Dotson flowline, and even more so for the Cosgrove flowline. For all the considered flowlines, the bed profiles have sections of retrograde slope susceptible of giving rise to a MISI, unless it can be prevented by sufficient lateral buttressing (Gudmundsson et al., 2012). A MISI seems to occur for the Cosgrove

and Dotson flowlines when the Budd law is used and for the PIG and Thwaites flowlines when the Schoof law is used. Note





that the ice shelves of Cosgrove and Dotson have completely disappeared at the end of EXP_ABMB with both the Schoof and Budd laws. On the contrary, whatever the chosen law, PIG and Thwaites ice shelves are conserved until the end of the experiment, although they get thinner.

## 4 Discussion

Among the three inferred states, $I_{R_{\gamma,100}}$ appears to be the most physically acceptable one, despite a RMS error slightly higher than the one obtained for $I_{R_{\gamma,1}}$. Indeed, for $I_{R_{\gamma,100}}$, the inferred viscosity has been reduced, at most, by $84\%$ (at the shear margins) and increased, at most, by $144\%$ (very locally in the higher part of the Kohler Glacier which feeds the Dotson ice shelve) compared to the reference viscosity $\eta_{0,ref}$. In contrast, for $I_{R_{\gamma,1}}$, the inferred viscosiy is at least 3 times higher than $\eta_{0,ref}$ on large parts of the domain and more than 15 times higher in some very localised areas close to the Kohler Glacier (regions in red in Fig. 4i). Without considering any of the other parameters affecting the ice viscosity, such an increase could only be explained by a reduction of the ice temperature of several tens of degrees celsius (up to $50$ °C in some regions) compared to the temperature map of Van Liefferinge and Pattyn (2013). On the other hand, even if ice was assumed to be at the melting point, the viscosity reduction observed in some other parts of the domain (regions in blue in Fig. 4i) is non-physical. Note that for $I_{R_{\gamma,100}}$, an increase of the ice temperature of a few degrees only relative to the solution of Van Liefferinge and Pattyn (2013) is sufficient to explain the local reduction of ice viscosity observed in some regions (much less extended than for $I_{R_{\gamma,1}}$) after the inversion, except at the shear margins where the low viscosity bands can be attributed to the presence of damaged ice. Finally, the difference between modelled and observed velocities, especially on the ice shelves, are too high for the inferred state $I_{SV}$.

The basal melting rate increase prescribed for the perturbation experiments EXP_ABMB induces a thinning of the ice shelves which reduces their buttressing effect and causes a retreat of the GL. The amplitude of this retreat gets larger as more weight is put on basal shear stress to the detriment of viscosity during initialisation. Indeed, adjusting viscosity during initialisation leads to low viscosity bands at shear margins which hamper the transfer of lateral stresses toward the interior of ice shelves and, from there, toward the ice streams which feed them. It follows that the model states for which viscosity is inferred are less sensitive to a reduction of the buttressing effect as the contribution of this effect to the initial global stress balance is less important than for $I_{SV}$.

The different friction laws show different sensitivity to velocity and effective pressure - from no explicit dependence on $N$ for the Weertman laws to an explicit dependence over the whole domain for the Budd law - which leads to different evolutions of basal shear stress as the geometry of the domain evolves in response to the perturbation. Therefore, it is not surprising that, overall, grounded ice area evolutions and VAF losses show more sensitivity to the chosen friction law when more weight is put on basal shear stress to the detriment of viscosity during initialisation, i.e. for $I_{SV}$ than for $I_{R_{\gamma,100}}$ or $I_{R_{\gamma,1}}$. The choice of the friction law affects not only the grounded ice area evolution, but also the evolution of ice thickness. Thus, in some cases, two friction laws lead to similar evolutions of the grounded ice area but to significantly different VAF losses (e.g. the linear and non-linear Weertman laws for $I_{SV}$, see panels a and d of Fig. 7). The Weertman laws systematically predict the lowest



VAF losses. In addition, the linear Weertman law always predicts less VAF loss than the non-linear Weertman law, which is in line with the results of Ritz et al. (2015) showing a highest contribution of the Antarctic ice sheet to SLR as the Weertman law tend toward a perfectly plastic law, i.e. $m \to 0$ in Eq. (5). The two Schoof laws lead to significantly higher VAF losses than the Weertman laws, with one exception: the non-linear Weertman law for the inferred state $I_{R_{\gamma,1}}$ which predicts a VAF

loss very close to the ones obtained with the two Schoof laws. This is likely due to the fact that, for $I_{R_{\gamma,1}}$, there are several regions of unexpectedly high viscosity located a few tens of km upstream of the initial GL (Fig. 4i). Once the GL has reached these regions, ice is stiffer and hampers further retreat, which explains the stabilisation of grounded ice area seen in Fig. 7c. As a consequence, the rate of VAF losses obtained with the two Schoof laws decreases, which also occurs with the non-linear Weertman law but later on so that, at the end of EXP_ABMB, the relative VAF losses predicted by the three laws are almost

identical. On the other hand, for the Schoof law, the lowest the value of $C_{max}$ the highest the predicted VAF losses. This is not surprising as the regions over which friction is governed by a Coulomb regime when using a Schoof law gets wider when the value of $C_{max}$ gets lower (Brondex et al., 2017). For $I_{R_{\gamma,100}}$, the VAF losses obtained with the Budd law are dramatically higher than the one obtained with the four other laws, which is in line with the results obtained in our previous study.

The sensitivity of ice thickness rates of change to the chosen friction law explains why the Budd law produces much larger

GL retreat than the four other laws in the regions of Cosgrove and Dotson and, conversely, less retreat in the Thwaites region. As shown in Brondex et al. (2017), because the Schoof law induces a Coulomb friction regime (i.e. $|\boldsymbol{\tau}_b| \sim C_{max}N$) within a narrow region of a few km right upstream the GL, the buttressing loss due to basal melting rate increase cannot be compensated within this region, despite an increase of velocities. Therefore, the perturbation is transmitted upstream, in areas governed by a Weertman friction regime (i.e. $|\boldsymbol{\tau}_b| \sim C_S|\mathbf{u}_b|^m$), where a velocity increase enables a rapid compensation of the buttressing loss.

As a consequence, although the Schoof law induces an important peak of thinning, it stays very localised in the immediate vicinity of the GL. In contrast, for the Budd law the basal shear stress depends on both $\mathbf{u}_b$ and $N$ over the whole domain. In addition, $N$ can be low on large distances upstream the GL, especially where the bed profile is retrograde and induces a rapid increase of the water column height as going further upstream, while the ice thickness does not increase as fast. As a consequence, the peak of thinning obtained at the GL with the Budd law is less pronounced than the one obtained with the

Schoof law, but, depending on the bed profile, it can propagate much further upstream. This is particularly well-illustrated in Fig. 3 of Brondex et al. (2017). In the regions of Cosgrove and Dotson, the initial surface profiles are close to the flotation altitudes over large distances upstream the GL (Fig. 9). In these two regions, the Budd law induces a rapid purge of the VAF over these large distances, leading to important retreat of the GL. This retreat might be enhanced by a potential MISI as the GL ends up on retrograde bed slopes. Note however that the Cosgrove and Dotson ice shelves being both well-confined, knowledge

of the bed slopes alone is not sufficient to predict the occurence of a MISI (Gudmundsson et al., 2012; Haseloff and Sergienko, 2018).

In contrast, Thwaites ice shelf is unconfined and, at the beginning of the experiment, the GL lies at the downstream bound of a section of retrograde bed slope (solid black line in bottom left panel of Fig. 9). The initial thinning produced by the Schoof law in the very close vicinity of the GL is sufficient to induce a retreat within the reverse slope area, where a MISI likely sets

up. On the contrary, the thinning peak produced by the Budd law at the GL is not sufficient to induce a retreat of the latter over



the 105 years of the experiment. Finally, the GL retreats obtained with the Budd and Schoof laws for PIG are more similar. This could be a consequence of the fact that the GL already lies on a retrograde slope at $t = 0$ a.

Although the differences in terms of GL dynamics and VAF losses obtained with the different friction laws tend to decrease as more weight is put on viscosity during inversion, they remain significant for the most physically acceptable inferred state,

$I_{R_{\gamma,100}}$. In particular, the commonly used linear Weertman law predicts about $9\,\mathrm{mm}$ SLE less VAF losses at the end of the 105 years than the Schoof law, which relies on a stronger physical basis.

The lack of a physically based subglacial hydrological model to compute effective pressure is the main limitation of the work presented here. Indeed, the water pressure calculated based on the assumption of perfect hydrological connectivity to the ocean might well be underestimated in some places as geothermal heat flux and frictional heating are known to cause basal melting

which is susceptible to build up water pressure at the ice/bed interface, especially in regions where the subglacial drainage system is inefficient (Colloni et al., 2018). In addition, beside basal shear stress, ice viscosity is not the only poorly constrained field. In particular, there are important uncertainties regarding the bed elevation. Recently, Nias et al. (2018) derived new bedrock elevation and ice thickness maps of the PIG using a method based on the principle of mass conservation and compared these maps to the Bedmap2 data: they found substantial differences between the two, in particular right upstream the 1996 GL

where a topographic rise has been removed with their new method. In addition, they ran the two obtained geometries forward in time to 50 years using different friction laws and showed that SLR projections are at least as sensitive to the accuracy of the bedrock topography and initial ice thickness as to the choice of the friction law.

## 5    Conclusion

The present study constitutes an extension of the sensibility analysis of GL dynamics and VAF loss predictions to the choice

of a friction law presented in Brondex et al. (2017). Whereas the latter was based on a synthetic 2D flowline case, here we consider a real-world application, the Amundsen Sea Embayment, and therefore, have to address specific problems. First of all, the basal shear stress is not known and has to be deduced from observations of ice flow surface velocities through inverse methods. In addition, other model parameters are uncertains and can require to be adjusted based on observations. Thus, when ice viscosity is not inferred but simply deduced from available ice temperature maps, discrepancies between modeled and

observed velocities are high, in particular within ice shelves. On the contrary, putting too much weight on viscosity to the detriment of basal shear stress during inversion lead to non-physical viscosity field. The inferred state $I_{R_{\gamma,100}}$ turns out to be the most physically acceptable of the constructed model states as it allows a good fit between observed and modelled velocities while showing a viscosity field closer to our expectations compared with $I_{R_{\gamma,1}}$.

Once the basal shear stress has been inferred, the friction coefficient distributions of the Weertman and Budd laws producing

the same basal shear stress field with these laws can easily be identified. In contrast, by construction the Schoof law induces an upper bound on the computed basal shear stress equals to the value of the parameter $C_{max}$. This latter parameter accounts for till deformation and the values it can take are partly constrained by laboratory experiments. As a result, the value of $C_S$ cannot be identified directly at nodes too close to flotation. This difficulty is overcome by interpolating (or extrapolating) the



value of $C_S$ at these nodes from the values at nodes where it can be directly identified. This procedure induces significant but very localised, in particular within ice shelves, discrepancies between the recomputed velocity field and the reference velocity field used for the identification.

The schematic perturbation experiments carried out following the identification step demonstrate a significant influence of the chosen friction law on GL dynamics and mass loss projections at a century timescale. In line with results obtained in our previous study, the VAF loss projections produced with the commonly used Weertman law are systematically lower than the ones obtained with the Schoof law, which themself are surpassed by the projection carried out with the Budd law (for $I_{R_{\gamma,100}}$). In addition, the Budd law being dependent on effective pressure over the whole grounded domain, it induces GL retreat in places where the other tested laws do not. Although the differences between the results produced with the various law tend to decrease as more weight is put on viscosity during inversion, they keep being significant for the most physically acceptable model state constructed, i.e. $I_{R_{\gamma,100}}$.

In light of these results, we conclude that no reliable projection of future sea level rise can be obtained without the use of a physically based friction law. Therefore, significant efforts still need to be put on getting a better understanding of the physical processes at play at the ice/bedrock interface in order to constain the form of the friction law which needs to be used in models. In particular, the recognized importance of water pressure for basal sliding must be explicitly accounted for through the use of an effective pressure-dependent law, but the latter ought to be fed by a physically based subglacial hydrological model.

*Code and data availability.* Elmer/Ice code is publicly available through GitHub (https://github.com/ElmerCSC/elmerfem). All the simulations were performed with the version 8.2 (Rev: 997cb45) of Elmer/Ice. All scripts used for simulations and post-treatment as well as model output are available upon request from authors. The data used are listed in the references, except the surface mass balance which is available upon request from authors.

*Author contributions.* Julien Brondex, Fabien Gillet-Chaulet and Olivier Gagliardini designed the study. Fabien Gillet-Chaulet collected the data and prepared the experimental setup. Julien Brondex conducted the inversions, the identifications and the prognostic simulations. Julien Brondex wrote the paper with contributions from all co-authors.

*Competing interests.* Olivier Gagliardini is a member of the editorial board of the journal. All other authors declare that they have no competing interests.

*Acknowledgements.* This work was funded by the French National Research Agency (ANR) through the TROIS-AS (ANR-15-CE01-0005-01) project. J.B., F.G.C. and O.G. are part of Labex OSUG@2020 (ANR10 LABX56). Most of the computational resources were provided by CINES through the gge6066 project. Some of the computations were performed using the Froggy platform of the CIMENT infrastruc-





ture (https://ciment.ujf-grenoble.fr), which is supported by the Rhône-Alpes region (GRANT CPER07-13 CIRA), the OSUG@2020 labex (reference ANR10 LABX56), and the Equip@Meso project (reference ANR-10-EQPX-29-01) of the programme Investissements d'Avenir supervised by the ANR.





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



**Figure 4.** Comparison of inferred states $I_{SV}$ (first column), $I_{R_{\gamma},100}$ (second column) and $I_{R_{\gamma},1}$ (third column). For each of the three inferred states, the first row - panels (a), (b) and (c) - shows the absolute difference between modelled $\mathbf{u}$ and observed $\mathbf{u}_{obs}$ velocities (m a$^{-1}$); the second row - panels (d), (e) and (f) - show the ratio of the norm of $\boldsymbol{\tau}_b$ relatively to the norm of $\boldsymbol{\tau}_d$; the third row shows the reference viscosity $\eta_{0,ref}$ (MPa a$^{1/3}$) for the inferred state $I_{SV}$ - panel (g) - as well as the relative difference between the reference viscosity and the inferred viscosity for the inferred states $I_{R_{\gamma},100}$ (second column) and $I_{R_{\gamma},1}$ - panels (h) and (i), respectively. Note the different color scales for panels (h) and (i).





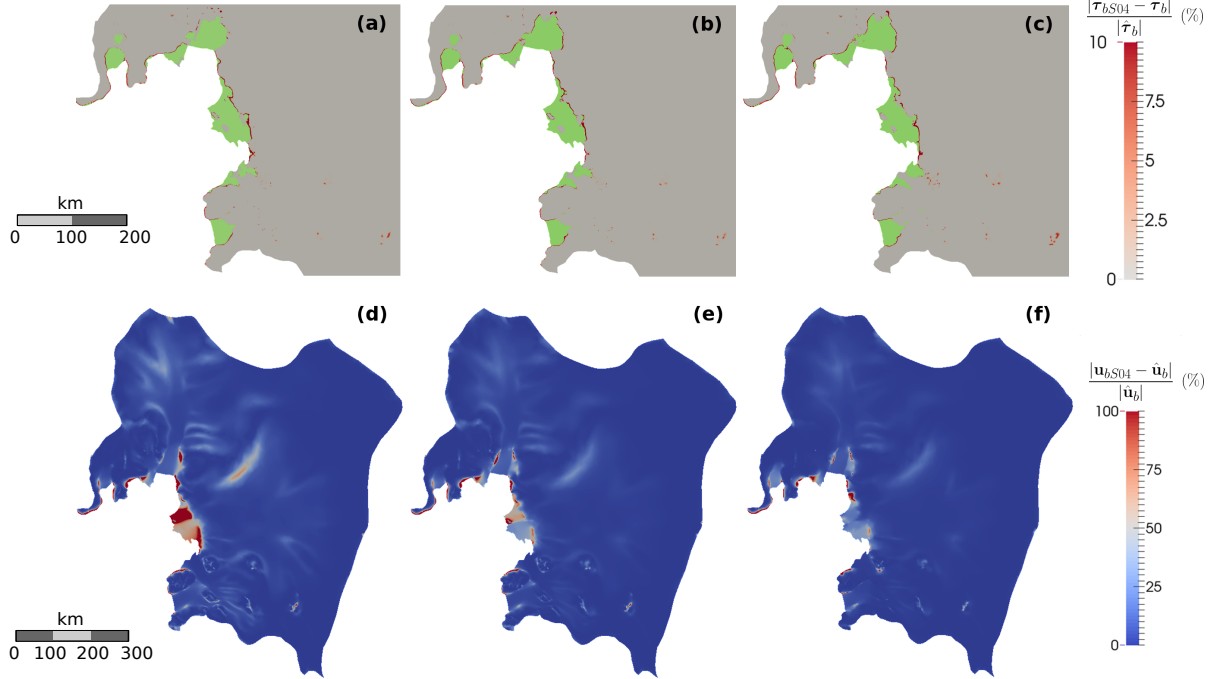

**Figure 5.** Relative differences (%) between the fields of $\boldsymbol{\tau}_b$ (first row) and $\mathbf{u}_b$ (second row) produced by the linear Weertman law and the non-linear Schoof law associated to $C_{max} = 0.4$ and to the $C_S$ distributions identified through Eq. (22) or interpolated, for the inferred states $I_{SV}$ - panels (a) and (d) -, $I_{R_{\gamma,100}}$ - panels (b) and (e) - and $I_{R_{\gamma,1}}$ - panels (c) and (f). Figures of the first row zoom in the neighborings of ice shelves (shown in green) where the relative differences in $\boldsymbol{\tau}_b$ are concentrated while figures of the second row show the whole ASE.

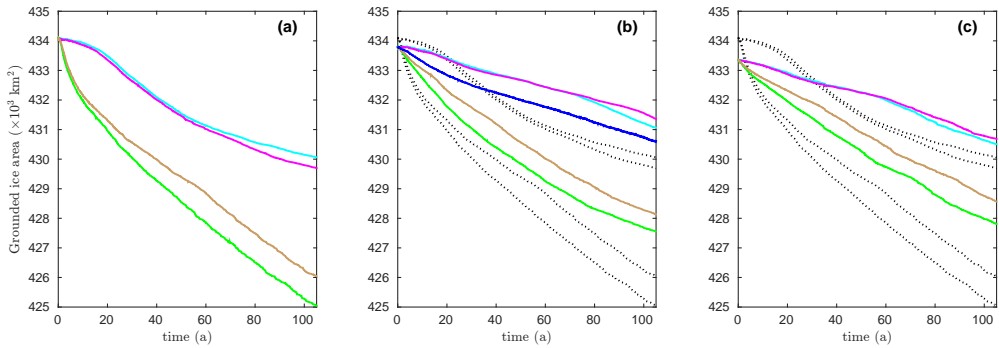

**Figure 6.** Evolution of grounded ice area ($\times 10^3$ km$^2$) as a function of time during EXP_CONTROL for the inferred states (a) $I_{SV}$, (b) $I_{R_{\gamma,100}}$ and (c) $I_{R_{\gamma,1}}$ and the friction laws linear Weertman (cyan), non-linear Weertman (magenta), Schoof with $C_{max} = 0.4$ (green), Schoof with $C_{max} = 0.6$ (brown) and non-linear Budd (blue, for the initial state $I_{R_{\gamma,100}}$ only). The results obtained with the various friction laws for the inferred state $I_{SV}$ are reported on (b) and (c) to ease comparison (black dotted lines).





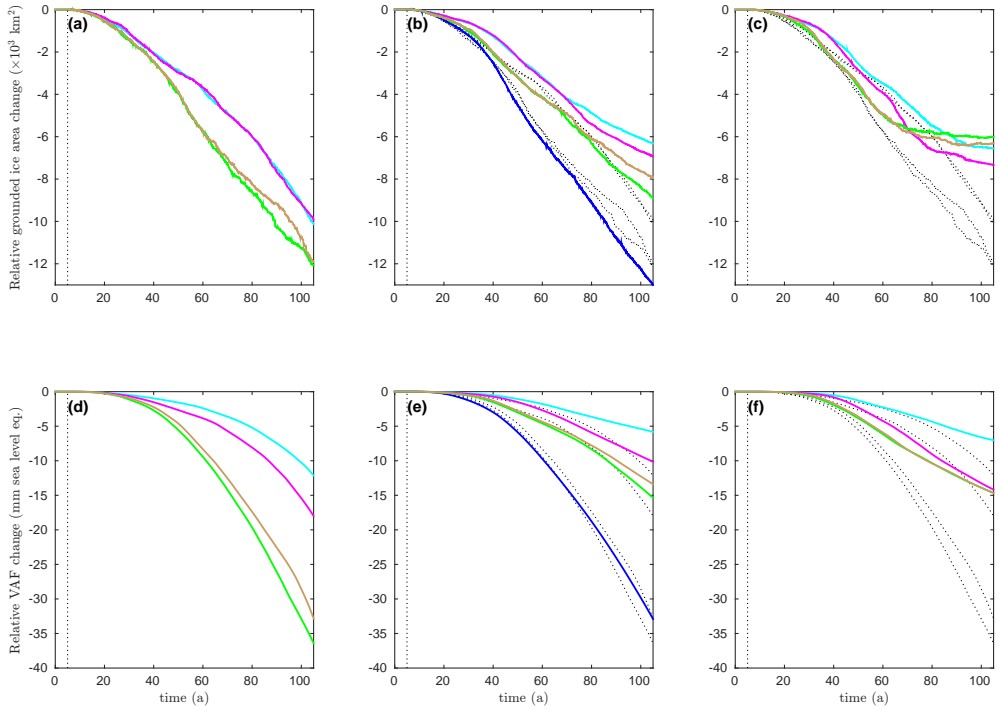

**Figure 7.** Evolution of grounded ice area in $\times 10^3$ km$^2$ (first row), and of volume above flotation in mm sea level equivalent (second row) as a function of time during EXP_ABMB (colored solid lines) relative to EXP_CONTROL for the inferred states $I_{SV}$ - panels (a) and (d) -, $I_{R_{\gamma,100}}$ - panels (b) and (e) - and $I_{R_{\gamma,1}}$ - panels (c) and (f) - with the friction laws linear Weertman (cyan), non-linear Weertman (magenta), Schoof with $C_{max} = 0.4$ (green), Schoof with $C_{max} = 0.6$ (brown) and non-linear Budd (blue, for the inferred state $I_{R_{\gamma,100}}$ only). The results obtained with the various friction laws for the inferred state $I_{SV}$ are reported on, respectively, (b)-(c) and (e)-(f) to ease comparison (black dotted lines). The vertical black dotted lines marks the introduction of the perturbation at $t = 5$ a.





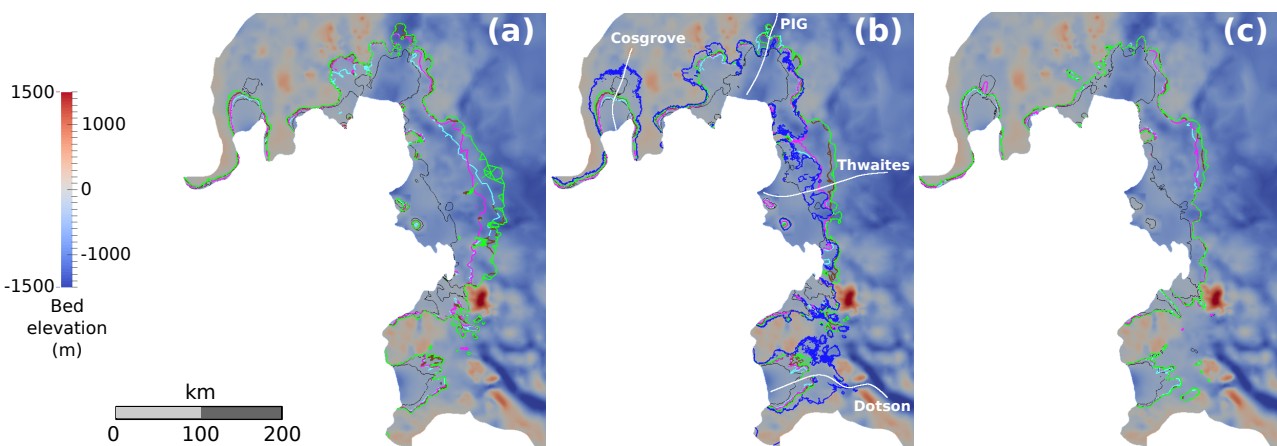

**Figure 8.** Bed elevation (m) and GL positions at the end of the experiement EXP_ABMB for the inferred states (a) $I_{SV}$, (b) $I_{R_{\gamma,100}}$ and (c) $I_{R_{\gamma,1}}$ and the friction laws linear Weertman (cyan), non-linear Weertman (magenta), Schoof with $C_{max} = 0.4$ (green), Schoof with $C_{max} = 0.6$ (brown) and non-linear Budd (blue, for the inferred state $I_{R_{\gamma,100}}$ only). The GL position at $t = 0$ a is reported for each of the three inferred states (solid black lines). The solid white lines in (b) correspond to flowlines at which ice-sheet profiles are represented in Fig. (9).



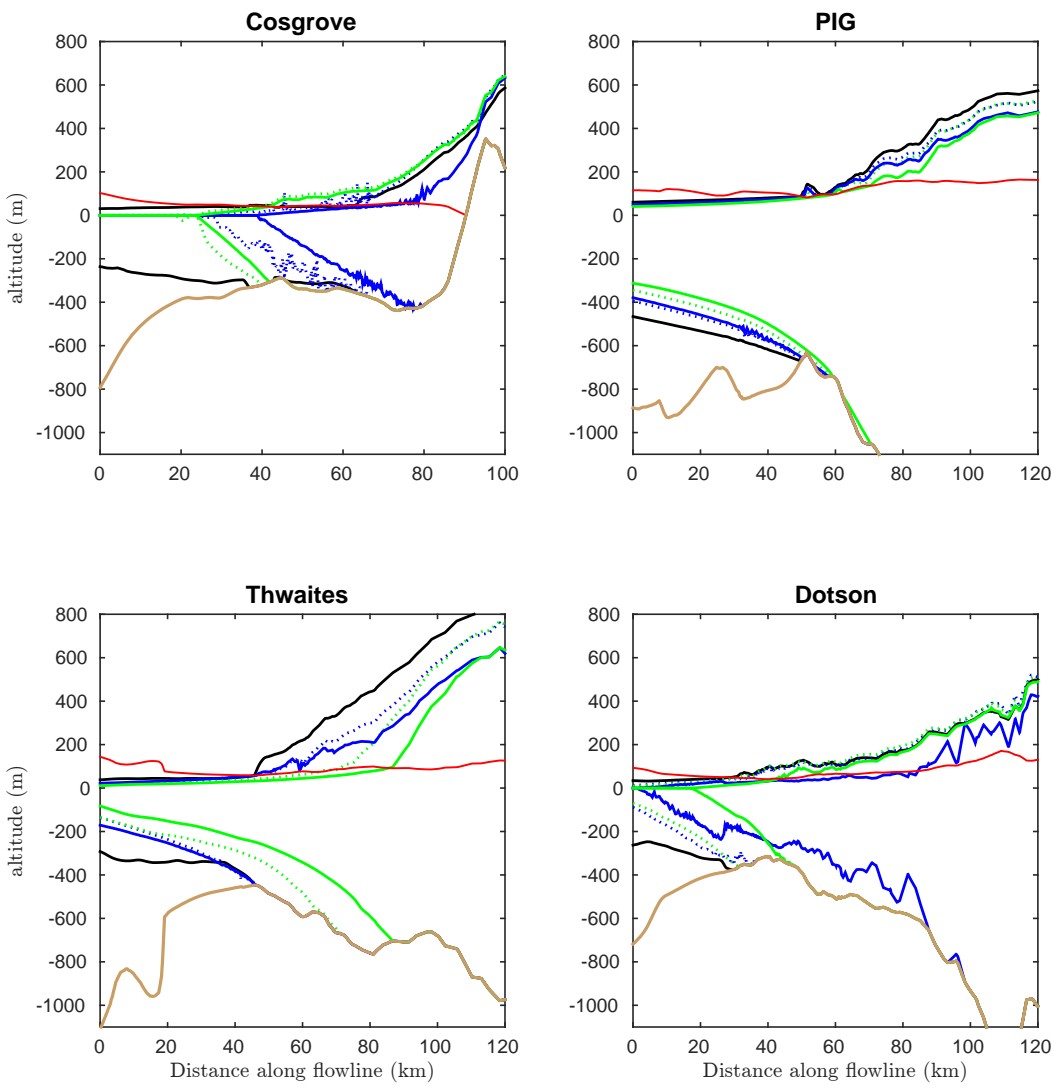

**Figure 9.** Ice-sheet profiles obtained for the inferred state $I_{R_{\gamma,100}}$ at $t = 0$ a (black solid line), $t = 55$ a (colored dotted line) and $t = 105$ a (colored solid line) of EXP_ABMB with the Schoof law associated to $C_{max} = 0.4$ (green) and the Budd law (blue), along the flowlines reported in Fig. 8. The solid light brown line is the bed elevation. The red solid line is the flotation altitude $z_f$, i.e. the altitude of the top surface for which the ice-sheet is exactly at flotation.