# Peer review of "Sensitivity of centennial mass loss projections of the Amundsen basin to the friction law"

_The Cryosphere, 2018_

## Referee Comment (RC1) · M. Morlighem (Referee) · 15 Oct 2018

The paper entitled "Sensitivity of centennial mass loss projections of the Amundsen basin to the friction law" by Julien Brondex, Fabien Gillet-Chaulet, and Olivier Gagliardini investigates the effect of 3 different basal friction laws (using different parameters) on projections of the Amundsen sea embayment over the next hundred years. The authors conducted a similar study last year but based on idealized geometry. They concluded that the evolution of the grounding line and volume above floatation over a 100-year simulation varies significantly depending on the choice of friction law used (Weertman, Budd or Schoof). Here, the authors conduct a similar investigation but applied to a real system, and extending this analysis to the Amundsen sea sector proved to be more difficult than with an idealized setting. It is shown that, for similar initial

states, the Weertman friction law systematically predicts less mass loss than any other law. The Budd friction law, on the other hand, induces more grounding line retreat and mass loss than the Schoof sliding law. The authors do not make recommendation on what friction law to use but put the finger on an important problem: ice sheet projections are strongly dependent on the choice of basal friction law and more research is necessary to better constrain basal friction.

**1 General Comments**

This is a very important topic that has not received a lot of attention so far. The paper is well written and easy to follow. I don't have any major comment but a few suggestions outlined below.

First, three friction laws are used:

- Linear Weertman ($m = 1$)

- Nonlinear Weertman ($m = 1/3$)

- Nonlinear Budd ($m = 1/3$)

- Nonlinear Schoof ($m = 1/3$ and $C_{max} = 0.4$)

- Nonlinear Schoof ($m = 1/3$ and $C_{max} = 0.6$)

I understand that the authors cannot run every possible combination, but it would be great if they could include the friction laws that are commonly used in the ice sheet modeling community. The linear Budd law ($m = 1$), for example, is extensively used by multiple groups (ISSM, UW, etc), much more than the nonlinear Budd law. It would be great to see how it compares with respect to other laws. These experiments could also

be useful for intercomparison projects such as ISMIP6 in order to better understand the differences in model projections, but this would only be possible if common laws are tested.

Also, why is the Budd sliding law only tested with one of the inferred initial states? Is it a problem of computational resources? It feels like the analysis is incomplete.

**2   Specific comments**

Lots of comma missing before "which" and "where".

- p1 l3: to a schematic perturbation → to a prescribed perturbation
- p1 l11: much higher → significantly higher
- p1 l21: trustworthy → reliable
- p1 l21: modelling of GL dynamics ...
- p1 l22: as close as possible to observations.
- p2 l1: have come up → have been developed
- p2 l7: have shown that, at large strain, the till ...
- p2 l13: behaves very similarly → behave similarly
- p2 l27: schematic perturbation → prescribed perturbation
- p2 l28: consider removing "To reach our goal,"
- p2 l33: we apply a synthetic perturbation to the basal melting rate under floating ice, to the different...

- p3 Figure 1: label Cosgrove ice shelf on the figure (in green)

- p3 eq 1: double check but given your definition of $\tau_b$, the sign in front of $\tau_{b,x}$ and $\tau_{b,y}$ should be $+$.

- p3 eq 2: Did you forget to divide the integral by the ice thickness $H$?

- p5 l6: I think $a_s$ should be defined as the surface mass balance instead of just the accumulation rate (it may be negative in some places). That's how it is defined in p6 l9

- p6 l12: these two pinning points ... to be critical because of the ...

- p6 l27: being submitted $\rightarrow$ undergoing

- p7 l18: impurity content

- p7 l18: the question is whether

- p9 l22: initializing the friction based on driving stress was first proposed by Morlighem et al. 2013 :)

- p10 l11: the coefficient of other laws, which ...

- p10 l30: all over the model domain

- p11 l14: rephrase "are submitted to", maybe "the 13 initial states are then run for 105 years under two different scenarios: (1) ..."

- p13 l17 seldom exceeds 10 % $\rightarrow$ rarely exceeds 10% (no space between before a percentage sign)

- p13 l24 significative $\rightarrow$ significant
- p13 l32: focus on → focuses on

- p14 l11: Independently → Irrespective

- p14 l16: all over the 105 a → over the entire 105 a

- p15 l32: as going further → as we go further

- p17 l10: the lower ... the higher

- p17 l18: an increase in ice velocities

- p17 l23: as going further → as we go further

- p17 l32: as Thwaites glacier is *mostly* unconfined

- p17 l35: a MISI likely *initiates*

- p19 l4: The perturbation experiments

- p19 l16: You have not really shown that a subglacial hydrology model is needed here, maybe rephrase the sentence

- p17 figure 8: increase the width of the grounding lines

**3 References**

M. Morlighem, H. Seroussi, E. Larour and E. Rignot, Inversion of basal friction in Antarctica using exact and incomplete adjoints of a higher-order model, Journal of Geophysical Research: Earth Surface 118 (2013) 1746–1753

---

## Referee Comment (RC2) · T. Kleiner (Referee) · 12 Nov 2018

**Review of**
**Sensitivity of centennial mass loss projections of the Amundsen basin to the friction law**
**by Brondex et al.**

Thomas Kleiner

**Summary**

It is very clear already from the early days of ice flow modelling, that the knowledge about subglacial conditions and the sliding relation is crucial for the understanding of ice sheet dynamics. This has been nicely elaborated in the previous study by Brondex et al. (2017) based on an idealised geometry set-up. In this study the authors conduct prognostic simulations for the Amundsen Sea Embayment (ASE) of about 100 years each to assess the relative sensitivity of grounded ice area and volume above flotation to the chosen friction law and initialisation strategy. The applied numerical flow model solves the two-dimensional shallow shelf approximation (SSA) for the stress balance. Model initial states are computed by means of a control method, where basal shear stresses and the viscosity are adjusted to minimise the misfit between modelled and observed surface velocities. A published data set of simulated ice sheet temperatures is used to account for the temperature dependency, initially. Within the prognostic simulations a synthetic perturbation of ice shelf basal melt rates is used to asses the model response to the given friction laws.

The paper is well written and well structured, and certainly deserves to be published. Nonetheless I would hope for a round of revisions to address my major concerns and the list of issues (below).

**Major concerns and suggestions**

Unfortunately, the authors do not conclude about the most appropriate friction low for the ASE study site. In addition to that, the discussion of the results is based on the numerical flow modelling only. Constraints on the friction law based on observational data are entirely ignored, but should be included to gain the scientific outcome of this study. The ASE is probably the area in Antarctica that is covered by observations the most (e.g. Rippin et al., 2011; Smith et al., 2013; Brisbourne et al., 2017). If discussed with respect to observations, the study could already be a significant contribution "on getting a better understanding of the physical processes at play at the ice/bedrock interface in order to constrain the form of the friction law which needs to be used in models."

I have overlooked this several times, but the rheology parameters $A_0$ and $Q$, given in Table 1, are very uncommon. Especially the pre-exponential factor $A_0$ for temperatures above -10° C

differs by several magnitudes from the commonly used Paterson and Budd (1982) parameterization of the Arrhenius Law Eq. (4). I have checked Elmer-Ice (Gillet-Chaulet et al., 2012), ISSM source code and PISM source code. They all use very similar values as in EISMINT (Payne et al., 2000). With the reported values in this study, the viscosity would be much larger (see Figure 1) and thus, the friction law more important. Given the specific ice rheology in

[Figure]

Figure 1: Rate factor $A$ as a function of the pressure corrected temperature ($T_{\mathrm{hom.}}$) and the effective viscosity $\eta(A, \tau_{\mathrm{e}})$ with the effective stress $\tau_{\mathrm{e}}$ for the Paterson and Budd (1982) rheology (blue) and Brondex et al. (red).

this study, I have strong doubts, whether the results can be transferred to other models. If these values are not just different (wrong?) in the table, I would highly recommend to re-run the experiments with a more common set of parameters.

**Detailed, line-by-line comments/suggestions**

**P. 1, L. 3:** "Amundsen Sea Embayement" → "Amundsen Sea Embayment"

**P. 1, L. 16:** "to the oceans" → "to the ocean"

**P. 1, L. 21:** "trustworthy" consider "accurate/reliable"

**P. 1, L. 22:** "subcentennial timescales" → "sub-centennial timescales"

**P. 1, L. 25:** "a long standing problem" → "a long-standing problem"

**P. 2, L. 19:** "geometry and velocity field" → "geometry and (the) surface velocity field"

**P. 2, L. 23:** " Adhalgeirsdottir et al. (2014) have shown ..."

**P. 2, L. 24:** Consider "initial state of the model" instead of "model initial state"

**P. 2, L. 27–28:** "Our work being based on a schematic perturbation scenario, the results ... of the ASE to SLR." This sentence appears to be incomplete.

**P. 3, L. 1?:** "two-dimensionnal" → "two-dimensional"

**P. 3, L. 1?:** Consider "shelfy-stream approximation (SSA)" instead of or in addition to "shallow shelf approximation (SSA)" here, because of the basal shear stresses. This is widely used in the literature for MacAyeal's equations (e.g. in Morlighem et al., 2010).

**P. 3, Eqns. (1,2):** Although formal correct I would recommend to rewrite Eq. 1 with $\bar{\eta}$ as the vertically averaged effective viscosity with units $\mathrm{Pa\,s}$ (instead of integrated; units: $\mathrm{Pa\,s\,m}$). Thus,

$$4\frac{\partial}{\partial x}\left(H\bar{\eta}\frac{\partial u}{\partial x}\right) + \ldots, \tag{1}$$

with

$$\bar{\eta} = \frac{1}{H}\int_{z_{\mathrm{b}}}^{z_{\mathrm{s}}} \ldots \, dz. \tag{2}$$

**P. 4, L. 1:** "$\ldots\eta_0$ is the viscosity given by $\ldots$"
It is very misleading to call $\eta_0$ a viscosity, because it is obviously not (see units, e.g. in your Fig. 4). The equations (2) and (3) are correct and also how they are applied is correct, but your $\eta_0$ is only a substitution for the temperature dependent contribution to the viscosity, thus

$$XXX = \frac{1}{2}A^{-1/n} = \frac{1}{2}B, \tag{3}$$

where $A$ is the rate factor depending on the temperature relative to the temperature melting point and $B$ is the associated rate factor (Greve and Blatter, 2009, p. 56). I am specifically asking for a better name and symbol for $XXX$.

**P. 4, L. 3:** Although $A$ is called "fluidity parameter" already in Brondex et al. (2017) consider to use the commonly used term "rate factor" instead (Gillet-Chaulet et al., 2012; Gagliardini et al., 2013). Consider to use $T'$ instead of $T$ to account for the different meaning. Please state clearly, if you have used the temperature or the pressure corrected temperature from Van Liefferinge and Pattyn (2013).

**P. 4, Eqns. (5–7):** The "-" signs in front of $\tau_{\mathrm{b,x}}$ and $\tau_{\mathrm{b,y}}$ appear to be wrong in Eq. (1) with this notation of the different friction laws. Consider to use $\tau_{\mathrm{b}} = \ldots$ as in Brondex et al. (2017, Eqns. (1–3)).

**P. 5, L. 6:** "where $a_{\mathrm{s}}$ is the meteoric accumulation rate applied to the top surface of the whole domain and $a_{\mathrm{b}}$ $\ldots$"
Use "surface mass balance" for $a_{\mathrm{s}}$ as on page 6 line 9. I would suggest something like "where $a_{\mathrm{s}}$ is the surface mass balance $a_{\mathrm{b}}$  $\ldots$". It should be stated that basal melt is ignored for the grounded part of the ice and why in another sentence.

**P. 5, Eq. (13):** "$\bar{\eta}$" $\rightarrow$ "$H\bar{\eta}$" with $\bar{\eta}$ being the average effective viscosity. See also P. 4, L. 1 above.

**P. 5, L. 25:** Although this can be guessed from the figures, it should be stated that the calving front is not evolving.

**P. 6, L. 30–33:** Why is the Budd law only applied to one of the inferred states?

**P. 6, L. 33:** "one of the inferred state" $\rightarrow$ "one of the inferred states"?

**P. 7, Table 1:** The numbers for the pre-exponential factors $A_0$ and and activation energies $Q$ for 'warm' and 'cold' ice are very unexpected. See the Major concerns and suggestions section.

**P. 7, L. 6:** "ice temperature map"
I am not sure what this means. The word map suggests something two-dimensional for me, but the temperature is used for the rate factor $A$ and thus $\eta_0$ within the integral of Eq. (2). May "three-dimensional temperature field/distribution" fits better.

If the temperature is a three-dimensional field, than it is not clear what is shown as map in your figure 4g.

**P. 7, L. 7:** "a reference  field" and rename $\eta_{0,\text{ref}}$ as mentioned above (P. 4, L. 1).

**P. 7, L. 6:** The temperature field from Van Liefferinge and Pattyn (2013), based on the model of Pattyn (2010) is a very important part for this study. Therefore, the methods used to get this field should be summarised within a few sentences. Which data set is applied here (ensemble mean, one specific ensemble member)?

**P. 7, L. 8:** "on each node of a regular grid"

**P. 7, L. 18:** "wether" → "whether"

**P. 8, L. 2–4:** "Indeed, several model states ... adjusting  the basal shear stress or  the viscosity."

**P. 8, L. 5:** "we construct three inferred states — denoted $I_{SV}$, $I_{R_{\gamma,100}}$ and $I_{R_{\gamma,1}}$ — by means of the control method"
At this place the inferred states are introduced by names and the reader needs to continue reading until page 9, line 17 for the explanation of $I_{R_{\gamma,100}}$ and $I_{R_{\gamma,1}}$. This might be unavoidable as a number of equations must be presented first. Nevertheless, I missed the explanation of the subscript 'SV' in $I_{SV}$ until the end of the document.

**P. 9, L. 3-4:** Consider to move "respectively" further to the end of the sentence: "... which are related to the linear Weertman law coefficient and the viscosity, respectively, as follows:", but this is personal preference only.

**P. 9, L. 32:** "occurence" → "occurrence"

**P. 10, L. 21-22:** "except for the Budd law for which the identification has been done only for the case $I_{R_{\gamma,1}}$"
Why?

**P. 10, L. 23:** "at every grounded node covered with ice"

**P. 10, L. 24:** "which are ice free" → "which are ice-free"

**P. 11, L. 5:** "which are ice free" → "which are ice-free"

**P. 12, L. 11:** "local adjuste of viscosity"

**P. 12, L. 18:** "the inversion algorithm"

**P. 12, L. 23:** "has already been showed" → "has already been shown"

**P. 12, L. 21–25:** "It is also this same mechanism ... Borstad et al., 2012, 2013)." Although damage could play a role, I am not convinced of this argument.

I think, the shear margins are just not well enough resolved in the velocity field that has been simulated in the study by Van Liefferinge and Pattyn (2013, 5 km horizontal resolution). Unfortunately, ice flow velocities are not presented in Van Liefferinge and

Pattyn (2013) or Pattyn (2010). The basal drag in an ice stream is usually low, thus the lateral drag at the shear margins balances the ice stream's driving stress. Similar to the condition at an ice sheets base, the drag leads to deformation of ice (strain) and thus strain heating. As the viscosity depends on temperature the viscosity decreases (see e.g. Bondzio et al., 2017). This is supported by your figure 4 panel g, where no viscosity variations across the shear margins of PIG near the GL are visible.

The cited literature is only related to 'damage' in ice shelves (Larsen B and C) and not appropriate for the conditions in the ASE.

**P. 13, L. 1:** "loosing" → "losing"

**P. 13, L. 10:** "at every grounded nodes"

**P. 13, L. 19:** "whithin" → "within"

**P. 13, L. 31:** "the relative differences on in the velocity field"?

**P. 14, L. 1:** "the gaussian integration" → "the Gaussian integration"

**P. 15, L. 19:** "is primary controlled by" → "is primarily controlled by"

**P. 15, L. 23:** "significantly different than the" → "significantly different from the"
My preference.

**P. 15, L. 23:** "$z_f = \ldots$, constitutes the thickness above flotation."
This is only true for grounded ice. Consider to show the flotation altitude (red line in Fig. 9) only for the grounded part.

**P. 16, L. 7:** "Dotson ice shelve" → "Dotson Ice Shelf"

**P. 16, L. 8:** "viscosiy" → "viscosity"

**P. 16, L. 11:** "tens of degrees celsius" → "tens of degrees Celsius"

**P. 16, L. 12:** "temperature map" See comment above (P. 7, L. 6).

**P. 17, L. 2:** "showing a highest contribution" → "showing the highest contribution"

**P. 17, L. 28:** "leading to important retreat of the GL" → "leading to an/the important retreat of the GL"

**P. 17, L. 30:** "occurence" → "occurrence"

**P. 17, L. 33:** "solid black line in bottom left panel of Fig. 9" → "solid black line in the bottom left panel of Fig. 9"

**P. 18, L. 23:** "parameters are uncertains" → "parameters are uncertain"

**P. 18, L. 24:** "viscosity is not inferred but simply deduced"

**P. 18, L. 24:** "ice temperature maps" See above.

**P. 18, L. 31:** "equals to the value of" or "is equal to the value of"

**P. 19, L. 1–2:** Consider to rearrange the sentence (personal preference only). E.g. "This procedure induces significant but very localised discrepancies between the recomputed velocity field and the reference velocity field used for the identification, in particular within ice shelves." or "...particularly within ice shelves."

**P. 19, L. 4–16:** The authors state very clear at the beginning (P. 2, L. 28), that "...the results presented here should not be considered as actual projections of the future contribution of the ASE to SLR." Consider to choose other terms to replace "projections" within this and other parts of the text.

**P. 19, L. 14:** "constain" → "constrain"

**P. 24, Fig. 4:** I think, maps of the basal shear stress $|\tau_b|$ are required in addition to the stress ratios presented in (d,e,f) for the three inferred states. This would allow to compare your inversion with other modelling studies conducted in this area (e.g. Joughin et al., 2009; Morlighem et al., 2010) and observational data.

A large portion of your model domain appears white in the panels a–c indicating that observed velocities are not available here. This is not so easy to see in Rignot et al. (2011), but in Mouginot et al. (2014, Fig. 1). Please explain how do you conduct the inversion in these areas. It is not clear, how the features in the panels d–f can be explained, given the extensive data gap in a–c.

**P. 25, Fig. 5:** I can't see any difference between a,b and c. The tiny little areas in between the green and grey areas appear all just red. The zoom in area should be marked in one of the figures for the whole ASE.

**P. 27, Fig. 8:** The coloured lines should be slightly thicker.

**References**

Bondzio, J. H., Morlighem, M., Seroussi, H., Kleiner, T., Rückamp, M., Mouginot, J., Moon, T., Larour, E. Y., and Humbert, A.: The mechanisms behind Jakobshavn Isbræ's acceleration and mass loss: A 3-D thermomechanical model study, Geophysical Research Letters, 44, 6252–6260, doi:10.1002/2017GL073309, 2017.

Brisbourne, A. M., Smith, A. M., Vaughan, D. G., King, E. C., Davies, D., Bingham, R. G., Smith, E. C., Nias, I. J., and Rosier, S. H. R.: Bed conditions of Pine Island Glacier, West Antarctica, Journal of Geophysical Research: Earth Surface, 122, 419–433, doi:10.1002/2016jf004033, 2017.

Brondex, J., Gagliardini, O., Gillet-chaulet, F., and Durand, G.: Sensitivity of grounding line dynamics to the choice of the friction law, Journal of Glaciology, 63, 854–866, doi:10.1017/jog.2017.51, 2017.

Gagliardini, O., Zwinger, T., Gillet-Chaulet, F., Durand, G., Favier, L., de Fleurian, B., Greve, R., Malinen, M., Martín, C., Råback, P., Ruokolainen, J., Sacchettini, M., Schäfer, M., Seddik, H., and Thies, J.: Capabilities and performance of Elmer/Ice, a new-generation ice sheet model, Geoscientific Model Development, 6, 1299–1318, doi:10.5194/gmd-6-1299-2013, 2013.

Gillet-Chaulet, F., Gagliardini, O., Seddik, H., Nodet, M., Durand, G., Ritz, C., Zwinger, T., Greve, R., and Vaughan, D. G.: Greenland ice sheet contribution to sea-level rise from a new-generation ice-sheet model, The Cryosphere, 6, 1561–1576, doi:10.5194/tc-6-1561-2012, 2012.

Greve, R. and Blatter, H.: Dynamics of Ice Sheets and Glaciers, Advances in Geophysical and Environmental Mechanics and Mathematics, Springer Berlin Heidelberg, doi:10.1007/978-3-642-03415-2, 2009.

Joughin, I., Tulaczyk, S., Bamber, J. L., Blankenship, D., Holt, J. W., Scambos, T., and Vaughan, D. G.: Basal conditions for Pine Island and Thwaites Glaciers, West Antarctica, determined using satellite and airborne data, Journal of Glaciology, 55, 245–257, doi:10.3189/002214309788608705, 2009.

Morlighem, M., Rignot, E., Seroussi, H., Larour, E., Ben Dhia, H., and Aubry, D.: Spatial patterns of basal drag inferred using control methods from a full-Stokes and simpler models for Pine Island Glacier, West Antarctica, Geophysical Research Letters, 37, L14 502, doi:10.1029/2010GL043853, 2010.

Mouginot, J., Rignot, E., and Scheuchl, B.: Sustained increase in ice discharge from the Amundsen Sea Embayment, West Antarctica, from 1973 to 2013, Geophysical Research Letters, 41, 1576–1584, doi:10.1002/2013gl059069, 2014.

Paterson, W. S. B. and Budd, W. F.: Flow parameters for ice sheet modelling, Cold Regions Science and Technology, 6, 175–177, doi:10.1016/0165-232X(82)90010-6, 1982.

Pattyn, F.: Antarctic subglacial conditions inferred from a hybrid ice sheet/ice stream model, Earth and Planetary Science Letters, 295, 451–461, doi:10.1016/j.epsl.2010.04.025, 2010.

Payne, A. J., Huybrechts, P., Abe-Ouchi, A., Calov, R., Fastook, J. L., Greve, R., Marshall, S. J., Marsiat, I., Ritz, C., Tarasov, L., and Thomassen, M. P. A.: Results from the EISMINT model intercomparison: the effects of thermomechanical coupling, Journal of Glaciology, 46, 227–238, 2000.

Rignot, E., Mouginot, J., and Scheuchl, B.: Ice Flow of the Antarctic Ice Sheet, Science, 333, 1427–1430, doi:10.1126/science.1208336, 2011.

Rippin, D., Vaughan, D., and Corr, H.: The basal roughness of Pine Island Glacier, West Antarctica, Journal of Glaciology, 57, 67–76, doi:10.3189/002214311795306574, 2011.

Smith, A. M., Jordan, T. A., Ferraccioli, F., and Bingham, R. G.: Influence of subglacial conditions on ice stream dynamics: Seismic and potential field data from Pine Island Glacier, West Antarctica, Journal of Geophysical Research: Solid Earth, 118, 1471–1482, doi:10.1029/2012jb009582, 2013.

Van Liefferinge, B. and Pattyn, F.: Using ice-flow models to evaluate potential sites of million year-old ice in Antarctica, Climate of the Past, 9, 2335–2345, doi:10.5194/cp-9-2335-2013, 2013.

---

## Author Comment (AC1) · 13 Dec 2018

**Referee #1: Mathieu Morlighem**

First, we would like to thank Mathieu Morlighem for his insightful comments on our paper.

**General comments**

This is a very important topic that has not received a lot of attention so far. The paper is well written and easy to follow. I don't have any major comment but a few suggestions outlined below.
First, three friction laws are used:

- Linear Weertman ($m = 1$)

- Nonlinear Weertman ($m = 1/3$)

- Nonlinear Budd ($m = 1/3$)

- Nonlinear Schoof ($m = 1/3$ and $C_{max} = 0.4$)

- Nonlinear Schoof ($m = 1/3$ and $C_{max} = 0.6$)

I understand that the authors cannot run every possible combination, but it would be great if they could include the friction laws that are commonly used in the ice sheet modeling community. The linear Budd law ($m = 1$), for example, is extensively used by multiple groups (ISSM, UW, etc), much more than the nonlinear Budd law. It would be great to see how it compares with respect to other laws. These experiments could also be useful for intercomparison projects such as ISMIP6 in order to better understand the differences in model projections, but this would only be possible if common laws are tested. Also, why is the Budd sliding law only tested with one of the inferred initial states? Is it a problem of computational resources? It feels like the analysis is incomplete.

We acknowledge that the linear Budd law is extensively used in the modelling community as well. Therefore, we have decided to test this law too, but for the inferred state $I_{R\gamma,100}$ only. Indeed, given the size of the domain and the mesh refinement required to capture the main flow features, we had to split the computational domain in 24 partitions and to affect 4 CPUs to each partition so that simulation durations were not too long. As a consequence, running a 100-year pronostic simulation for one particular combination inferred state/friction law/pronostic experiment (i.e. ABMB or CONTROL) costs around 11000 hours CPU. Since the Budd law induces a depence of $|\boldsymbol{\tau}_b|$ on $N$ over the whole ice sheet, even in its interior where a drainage system may not be present, and since it does not imply any upper bound on the value of $|\boldsymbol{\tau}_b|/N$, we consider the Budd law as being the least physically acceptable one among all the tested laws. This is the reason why we have decided that both the linear and non-linear Budd laws would be tested for $I_{R\gamma,100}$ only, which is the most physically acceptable inferred state. Information regarding the computational cost of transient simulations, as well as explanations regarding the reason why the Budd laws are tested for $I_{R\gamma,100}$ only, have been added to the manuscript.

**Specific comments**

Lots of comma missing before "which" and "where".

Commas have been added before some of the "which" and "where".

- p1 l3: to a schematic perturbation $\rightarrow$ to a prescribed perturbation

- p1 l11: much higher $\rightarrow$ significantly higher

- p1 l21: trustworthy $\rightarrow$ reliable

- p1 l21: modelling of GL dynamics ...

- p1 l22: as close as possible to observations.

- p2 l1: have come up $\rightarrow$ have been developed

- p2 l7: have shown that, at large strain, the till ...

- p2 l13: behaves very similarly → behave similarly

- p2 l27: schematic perturbation → prescribed perturbation

- p2 l28: consider removing "To reach our goal,"

- p2 l33: we apply a synthetic perturbation to the basal melting rate under floating ice, to the different...

We followed your suggestions for all the points listed above.

- p3 Figure 1: label Cosgrove ice shelf on the figure (in green)

Figure 1 has been modified to add the name of all ice shelves (in black).

- p3 eq 1: double check but given your definition of $\tau_b$, the sign in front of $\tau_{b,x}$ and $\tau_{b,y}$ should be +.

You are right, this was a mistake, which has been corrected.

- p3 eq 2: Did you forget to divide the integral by the ice thickness H?

We think that the way $\bar{\eta}$ was defined in the first version of the manuscript was consistent with the way it was used in Eqs. (1) and (13). However, given your comment and the one of Referee #2, we have decided to redefine $\bar{\eta}$ as being the vertically averaged effective viscosity, which now reads $\bar{\eta} = \bar{\eta}_0 D_e^{(1-n)/n}$, with $\bar{\eta}_0 = \frac{1}{H} \int_{z_b}^{z_s} \frac{1}{2} A^{-1/n} dz$. Eqs. (1) and (13) have been corrected accordingly.

- p5 l6: I think $a_s$ should be defined as the surface mass balance instead of just the accumulation rate (it may be negative in some places). That's how it is defined in p6 l9

- p6 l12: these two pinning points ... to be critical because of the ...

- p6 l27: being submitted → undergoing

- p7 l18: impurity content

- p7 l18: the question is whether

We followed your suggestions for all the points listed above.

- p9 l22: initializing the friction based on driving stress was first proposed by Morlighem et al. 2013

The reference has been corrected.

- p10 l11: the coefficient of other laws, which ...

- p10 l30: all over the model domain

- p11 l14: rephrase "are submitted to", maybe "the 13 initial states are then run for 105 years under two different scenarios: (1) ..."

We followed your recommendations for all the points listed above.

- p13 l17 seldom exceeds 10 % → rarely exceeds 10% (no space between before a percentage sign)

This has been corrected and we have also removed all the spaces before percentage signs.

- p13 l24 significative → significant

- p13 l32: focus on → focuses on

- p14 l11: Independently → Irrespective

- p14 l16: all over the 105 a → over the entire 105 a

- p15 l32: as going further → as we go further

- p17 l10: the lower ... the higher

- p17 l18: an increase in ice velocities

- p17 l23: as going further $\rightarrow$ as we go further

- p17 l32: as Thwaites glacier is mostly unconfined

- p17 l35: a MISI likely initiates

- p19 l4: The perturbation experiments

We followed your suggestions for all the points listed above.

- p19 l16: You have not really shown that a subglacial hydrology model is needed here, maybe rephrase the sentence

The sentence has been modified.

- p17 figure 8: increase the width of the grounding lines

This has been done.

---

## Author Comment (AC2) · 13 Dec 2018

**Referee #2: Thomas Kleiner**

First, we would like to thank Thomas Kleiner for his insightful comments on our paper.

**General comments**

Unfortunately, the authors do not conclude about the most appropriate friction law for the ASE study site. In addition to that, the discussion of the results is based on the numerical flow modelling only. Constraints on the friction law based on observational data are entirely ignored, but should be included to gain the scientific outcome of this study. The ASE is probably the area in Antarctica that is covered by observations the most (e.g. Rippin et al., 2011; Smith et al., 2013; Brisbourne et al., 2017). If discussed with respect to observations, the study could already be a significant contribution "on getting a better understanding of the physical processes at play at the ice/bedrock interface in order to constrain the form of the friction law which needs to be used in models."

The scientific outcome of the present study is to unequivocally show that projections regarding the future dynamical contribution of an ice sheet to SLR are highly sensitive to the chosen friction law, even on timescales as short as 100 years. Using observations to constrain the form of the friction law which is best suited for a given application would surely constitute a huge leap towards producing reliable projections of future SLR, but, as explicitly stated in the introduction, is not the objective in the present study. It is true that several set of observations regarding the conditions beneath part of the ASE - usually limited to Pine Island as in the studies you cited - are available. However, for a given unique set of observations, the inferred basal stress must satisfy the global stress balance, so the solution of the inverse problem leads to the same basal stress whatever the chosen friction law. Constraining the form of the friction law would then require observations at different times with significant differences in basal velocities, basal stresses and water pressure at the ice/bed interface. Unfortunately, these multiple sets of observations are not available or incomplete. In particular, the water pressure at the ice/bed interface is largely unknown and the assumption of perfect hydrological connection to the ocean is too gross for the purpose of constraining the form of the friction law. In addition, although the dependence of basal shear stress to basal roughness is well acknowledged, formulating a mathematical relationship relating parameters regarding the topographical properties of the bed (local roughness, maximal slope of bumps,...) to the basal shear stress is far from being straightforward.
For all these reasons, all the friction laws commonly used in modelling studies are derived from theoretical arguments (e.g Weertman, 1957; Schoof, 2005; Tsai et al., 2015) or from laboratory experiments (e.g Budd et al., 1979; Iverson et al., 1998) but none of them has been validated *in situ*. Since they report the presence of soft sediments beneath Pine Island Glacier, the studies of Smith et al. (2013) and Brisbourne et al. (2017) tend to support, in this region only, the use of a Schoof law, rather than a Weertman or a Budd law, as the former induces a Coulomb friction regime in the vicinity of the GL, which has been shown from laboratory experiments to be the most adapted friction regime to represent the deformation of sediments (e.g. Iverson et al., 1998). Yet, this is in no way a validation of the Schoof law, and it does not provide any information regarding the spatial distribution which should be adopted for the parameters $C_S$, $C_{max}$ and $n$.
These aspects have been briefly discussed in Brondex et al. (2017), and a bit more thoroughly in Gillet-Chaulet et al. (2016). Anyway, we have added a small paragraph within the discussion section to state that, although available observations are not sufficient to constrain the form of the friction law which ought to be used, they tend to support the use of a Schoof law rather than the two other laws.

I have overlooked this several times, but the rheology parameters $A_0$ and $Q$, given in Table 1, are very uncommon. Especially the pre-exponential factor $A_0$ for temperatures above $-10°$ C differs by several magnitudes from the commonly used Paterson and Budd (1982) parameterization of the Arrhenius Law Eq. (4). I have checked Elmer-Ice (Gillet-Chaulet et al., 2012), ISSM source code and PISM source code. They all use very similar values as in EISMINT (Payne et al., 2000). With the reported values in this study, the viscosity would be much larger (see Figure 1) and thus, the friction law more important. Given the specific ice rheology in this study, I have strong doubts, whether the results can be transferred to other models. If these values are not just different (wrong?) in the table, I would highly recommend to re-run the experiments with a more common set of parameters.

The rheology parameters used in our study are calibrated from values recommended by Cuffey and Paterson (2010). For $T < -10°$ C, these authors suggest to use $Q_1 = 60$ kj mol$^{-1}$, which was already the value suggested by Paterson and Budd (1982). For $T > -10°$ C, Cuffey and Paterson (2010) suggest to use $Q_2 = 115$ kj mol$^{-1}$, whereas Paterson and Budd (1982) recommended $Q_2 = 139$ kj mol$^{-1}$. In addition, according to Cuffey and

Paterson (2010), the value of the rate factor at $T = -10°$ C should be $A(-10°$ C$) = 3.5 \times 10^{-25}$ Pa$^{-3}$s$^{-1}$ rather than $A(-10°$ C$) = 4.4 \times 10^{-25}$ Pa$^{-3}$s$^{-1}$ as recommended by Paterson and Budd (1982). These differences in the values of $Q_2$ and $A(-10°$ C$)$ explain the difference of several order of magnitudes in the value of the pre-exponential factor $A_0(T > -10°$ C$)$ between the one we used (Table 1 of the manuscript) and the one suggested by Paterson and Budd (1982). Note however that, as it can be seen on Fig. 1 of your review, the rate factor $A$ used in our study and the one deduced from the values of Paterson and Budd (1982) differ at most by a factor of 2 (i.e. when $T \to 0°$ C), despite the gap in $A_0(T > -10°$ C$)$. Note also that, although the values of Paterson and Budd (1982) were the ones adopted by most of the authors a few years ago (e.g Payne et al., 2000; Winkelmann et al., 2011; Larour et al., 2012; Gillet-Chaulet et al., 2012), many recent modelling studies, including studies based on ISSM (e.g. Morlighem et al., 2016; Yu et al., 2016; Seroussi et al., 2017) or Elmer/Ice (e.g. Gillet-Chaulet et al., 2016), make use of the new values recommended by Cuffey and Paterson (2010). In addition, in many studies an enhancement factor is included in order to modify the viscosity. In such a case, it makes no sense to compare the rate factors without including the effect of the enhancement factor, which often differs from one study to the other.
For all these reasons, we did not consider running the experiments with other rheology parameters. We have simply added the reference to Cuffey and Paterson (2010).

**Specific comments**

- P. 1, L. 3: "Amundsen Sea Embayement" $\to$ "Amundsen Sea Embayment"

- P. 1, L. 16: "to the oceans" $\to$ "to the ocean"

- P. 1, L. 21: "trustworthy" consider "accurate/reliable"

- P. 1, L. 22: "subcentennial timescales" $\to$ "sub-centennial timescales"

- P. 1, L. 25: "a long standing problem" $\to$ "a long-standing problem"

- P. 2, L. 19: "geometry and velocity field" $\to$ "geometry and (the) surface velocity field"

- P. 2, L. 23: "Yet, Adhalgeirsdottir et al. (2014) have shown . . . "

- P. 2, L. 24: Consider "initial state of the model" instead of "model initial state"

We followed your suggestions for all the points listed above.

- "Our work being based on a schematic perturbation scenario, the results ... of the ASE to SLR." This sentence appears to be incomplete.

We think that this sentence is actually complete.

- P. 3, L. 1?: "two-dimensionnal" $\to$ "two-dimensional"

It has been corrected.

- P. 3, L. 1?: Consider "shelfy-stream approximation (SSA)" instead of or in addition to "shallow shelf approximation (SSA)" here, because of the basal shear stresses. This is widely used in the literature for MacAyeal's equations (e.g. in Morlighem et al., 2010).

We followed your recommendation.

- Although formal correct I would recommend to rewrite Eq. 1 with $\bar{\eta}$ as the vertically averaged effective viscosity with units Pa s (instead of integrated units: Pa s m). Thus, ...

We followed your recommendation.

- P. 4, L. 1: "... $\eta_0$ is the viscosity given by ..." It is very misleading to call $\eta_0$ a viscosity, because it is obviously not (see units, e.g. in your Fig. 4). The equations (2) and (3) are correct and also how they are applied is correct, but your $\eta_0$ is only a substitution for the temperature dependent contribution to the viscosity, thus

$$XXX = \frac{1}{2}A^{-1/n} = \frac{1}{2}B, \tag{1}$$

where A is the rate factor depending on the temperature relative to the temperature melting point and B is the associated rate factor (Greve and Blatter, 2009, p. 56). I am specifically asking for a better name and symbol for XXX.

It is true that $\eta_0$ is not, properly speaking, a viscosity. However, since adjusting this quantity is an indirect way to adjust effective viscosity itself, we think it will be clearer for the reader if we keep the notation as it was in the previous version of the manuscript. Therefore, we have simply replaced $\eta_0$ by $\bar{\eta}_0$, as well as $\eta_{0,ref}$ by $\bar{\eta}_{0,ref}$, in order to stress that these quantities are vertical averages. In addition, we have modified the manuscript so that these quantities are no more referred to as "viscosities" in the text.

- P. 4, L. 3: Although A is called "fluidity parameter" already in Brondex et al. (2017) consider to use the commonly used term "rate factor" instead (Gillet-Chaulet et al., 2012; Gagliardini et al., 2013). Consider to use $T'$ instead of $T$ to account for the different meaning. Please state clearly, if you have used the temperature or the pressure corrected temperature from Van Liefferinge and Pattyn (2013).

We have changed "fluidity parameter" for "rate factor" as suggested. We have also changed $T$ for $T'$ and explicited the fact that $T'$ is the temperature relative to the pressure-melting point.

- P. 4, Eqns. (5-7): The "-" signs in front of $\tau_{b,x}$ and $\tau_{b,y}$ appear to be wrong in Eq. (1) with this notation of the different friction laws. Consider to use $\tau_b = ...$ as in Brondex et al. (2017, Eqns. (1-3)).

The "-" signs in Eq. (1) were indeed wrong and have been corrected. We decided to keep the form of Eqs (5-7) as in the previous version of the manuscript, as we want to stress the fact that, in the present study, $\boldsymbol{\tau}_b$ and $\mathbf{u}_b$ are vectors, which are aligned and with opposite directions. In Brondex et al. (2017), the geometry was unidimensional and we could directly write $\tau_b = ....$

- P. 5, L. 6: "where $a_s$ is the meteoric accumulation rate applied to the top surface of the whole domain and $a_b$ ..." Use "surface mass balance" for $a_s$ as on page 6 line 9. I would suggest something like "where $a_s$ is the surface mass balance $a_b$ applied to the top surface of the whole domain ...". It should be stated that basal melt is ignored for the grounded part of the ice and why in another sentence.

We have changed "meteoric accumulation rate" for "surface mass balance". We have also added "Basal melt at the ice/bed interface is neglected."

- P. 5, Eq. (13): "$\bar{\eta}$" $\rightarrow$ "$H\bar{\eta}$" with $\bar{\eta}$ being the average effective viscosity. See also P. 4, L. 1 above.

This has been changed.

- P. 5, L. 25: Although this can be guessed from the figures, it should be stated that the calving front is not evolving.

This information has been added.

- P. 6, L. 30-33: Why is the Budd law only applied to one of the inferred states?

See the answer to the general comments of Referee #1.

- P. 6, L. 33: "one of the inferred state" $\rightarrow$ "one of the inferred states"?

This has been corrected.

- P. 7, Table 1: The numbers for the pre-exponential factors $A_0$ and and activation energies $Q$ for 'warm' and 'cold' ice are very unexpected. See the Major concerns and suggestions section.

See the answer in the previous part.

- P. 7, L. 6: "ice temperature map" I am not sure what this means. The word map suggests something two-dimensional for me, but the temperature is used for the rate factor $A$ and thus $\eta_0$ within the integral of Eq. (2). May "three-dimensional temperature field/distribution" fits better. If the temperature is a three-dimensional field, than it is not clear what is shown as map in your figure 4g.

You are right, it is actually a 3D temperature field, which is used to derive a 3D field of $A$ based on Eq. (4), which is then vertically averaged to get $\bar{\eta}_{0,ref}$ based on Eq. (3). It is this vertical average that is shown in Fig. 4g. This has been made clearer in the manuscript.

- P. 7, L. 7: "a reference  field" and rename $\eta_{0,ref}$ as mentioned above (P. 4, L. 1).

This has been done. See the answer above.

- P. 7, L. 6: The temperature field from Van Liefferinge and Pattyn (2013), based on the model of Pattyn (2010) is a very important part for this study. Therefore, the methods used to get this field should be summarised within a few sentences. Which data set is applied here (ensemble mean, one specific ensemble member)?

In the present study we are using a number of datasets which are all equally important to construct our model initial states. All these datasets are correctly referenced and the interrested reader is free to read the corresponding articles if needed, including the paper of Van Liefferinge and Pattyn (2013). The temperature field that we have been using for the present study was actually provided by Van Liefferinge (personal communication), who made a specific run for this purpose as ice shelves were not included in the original work of Van Liefferinge and Pattyn (2013). This is now explicitly stated in the text.

- P. 7, L. 8: "on each node of a regular grid"

- P. 7, L. 18: "wether" → "whether"

These two mistakes have been corrected

- P. 8, L. 2-4: "Indeed, several model states ... adjusting rather the basal shear stress or rather the viscosity."

We have decided to leave this sentence as it was because we want to stress the fact that both the basal shear stress field and the viscosity field are adjusted at the same time, but with various relative weight.

- P. 8, L. 5: "we construct three inferred states - denoted $I_{SV}$ , $I_{R\gamma,100}$ and $I_{R\gamma,1}$ - by means of the control method" At this place the inferred states are introduced by names and the reader needs to continue reading until page 9, line 17 for the explanation of $I_{R\gamma,100}$ and $I_{R\gamma,1}$ . This might be unavoidable as a number of equations must be presented first. Nevertheless, I missed the explanation of the subscript 'SV' in $I_{SV}$ until the end of the document.

It is true that the reason why we use the notations $R\gamma, 100$ and $R\gamma, 1$ becomes obvious only from Eq. (19) or even a bit further. However, as you said, we have no choice as a number of equations must be presented first. It is also true that the subscribe $SV$ does not necessarily make sense in english and we have deciced to change it for $R\gamma, \infty$

- P. 9, L. 3-4: Consider to move "respectively" further to the end of the sentence: "... which are related to the linear Weertman law coefficient and the viscosity, respectively, as follows:", but this is personal preference only.

We followed your suggestion.

- P. 9, L. 32: "occurence" → "occurrence"

This has been corrected.

- P. 10, L. 21-22: "except for the Budd law for which the identification has been done only for the case $I_{R\gamma,100}$ " Why?

See the answer to the general comments of Referee #1.

- P. 10, L. 23: "at every grounded node covered with ice"

- P. 10, L. 24: "which are ice free" → "which are ice-free"

- P. 11, L. 5: "which are ice free" → "which are ice-free"

- P. 12, L. 11: "local adjustement of viscosity"

- P. 12, L. 18: "the inversion algorithm"

- P. 12, L. 23: "has already been showed" → "has already been shown"

All these points have been corrected.

- P. 12, L. 21-25: "It is also this same mechanism ... Borstad et al., 2012, 2013)." Although damage could play a role, I am not convinced of this argument. I think, the shear margins are just not well enough resolved in the velocity field that has been simulated in the study by Van Liefferinge and Pattyn (2013, 5 km horizontal resolution). Unfortunately, ice flow velocities are not presented in Van Liefferinge and Pattyn (2013) or Pattyn (2010). The basal drag in an ice stream is usually low, thus the lateral drag at the shear margins balances the ice stream's driving stress. Similar to the condition at an ice sheets base, the drag leads to deformation of ice (strain) and thus strain heating. As the viscosity depends on temperature the viscosity decreases (see e.g. Bondzio et al., 2017). This is supported by your figure 4 panel g, where no viscosity variations across the shear margins of PIG near the GL are visible. The cited literature is only related to 'damage' in ice shelves (Larsen B and C) and not appropriate for the conditions in the ASE.

The low viscosity bands to which we refer are located on Pine Island and Thwaites ice shelves and not within the grounded part of the ice streams (see Fig. 4h), therefore we think that the cited literature is totally relevant in this case. We agree that lateral drag at the shear margins must balance the driving stress as ice shelves do not experiment any basal drag. Beside strain heating which you are mentioning, it also causes locally high shear stresses leading to opening of fractures which makes ice softer. Indeed, several authors have reported the good correlation between these low viscosity bands and aerial observations of crevasses (e.g Borstad et al., 2013). Another mechanism which also tends to decrease ice viscosity at shear margins is the development of crystalline fabric which induces anisotropy in ice rheology, making ice softer in some stress directions and stiffer in others (Minchew et al., 2018). Saying whether the low viscosity bands are due to strain heating, anisotropy, damage, or to a combination of the three, is difficult. Therefore, we have added to the manuscript the fact that strain heating and anisotropy could also be potential explanations for the presence of these soft bands.

- P. 13, L. 1: "loosing" → "losing"

- P. 13, L. 10: "at every grounded nodes"

- P. 13, L. 19: "whithin" → "within"

- P. 13, L. 31: "the relative differences on in the velocity field"?

- P. 14, L. 1: "the gaussian integration" → "the Gaussian integration"

- P. 15, L. 19: "is primary controlled by" → "is primarily controlled by"

- P. 15, L. 23: "significantly different than the" → "significantly different from the" My preference.

We followed your suggestions for all the points listed above.

- P. 15, L. 23: "$z_f = ...$ , constitutes the thickness above flotation." This is only true for grounded ice. Consider to show the flotation altitude (red line in Fig. 9) only for the grounded part.

We have made this point clearer in the text as well as in the caption of Fig. 9. However, we chose not to change the latter as different "grounded parts" are actually represented in each plot, i.e. two for the Budd law, two for the Schoof law and the inital profile which is common to the two laws.

- P. 16, L. 7: "Dotson ice shelve" → "Dotson Ice Shelf"

- P. 16, L. 8: "viscosiy" → "viscosity"

- P. 16, L. 11: "tens of degrees celsius" → "tens of degrees Celsius"

We have corrected these points.

- P. 16, L. 12: "temperature map" See comment above (P. 7, L. 6).

The word "map" has been replaced by "field"

- P. 17, L. 2: "showing a highest contribution" → "showing the highest contribution"

- P. 17, L. 28: "leading to important retreat of the GL" → "leading to an/the important retreat of the GL"

- P. 17, L. 30: "occurence" → "occurrence"

- P. 17, L. 33: "solid black line in bottom left panel of Fig. 9" → "solid black line in the bottom left panel of Fig. 9"

- P. 18, L. 23: "parameters are uncertains" → "parameters are uncertain"

- P. 18, L. 24: "viscosity is not inferred but  deduced"

We followed your suggestions for all the points listed above.

- P. 18, L. 24: "ice temperature maps" See above.

The word "maps" has been replaced by "fields"

- P. 18, L. 31: "equals to the value of" or "is equal to the value of"

- P. 19, L. 1-2: Consider to rearrange the sentence (personal preference only). E.g. "This procedure induces significant but very localised discrepancies between the recomputed velocity field and the reference velocity field used for the identification, in particular within ice shelves." or "... particularly within ice shelves."

We followed your suggestions for the two points listed above.

- P. 19, L. 4-16: The authors state very clear at the beginning (P. 2, L. 28), that "... the results presented here should not be considered as actual projections of the future contribution of the ASE to SLR." Consider to choose other terms to replace "projections" within this and other parts of the text.

Although we do not make actual projections of the future contribution of the ASE to SLR in the present study, the latter shows sensivity of mass loss projections to the friction law and initialisation strategy. In this sense, the use of the word "projection" appears correct to us in most of the text. Yet, we agree that this term was unproperly used in one of the sentences of the conclusion section. This has been corrected.

- P. 19, L. 14: "constain" → "constrain"

This has been corrected.

- P. 24, Fig. 4: I think, maps of the basal shear stress $|\tau_b|$ are required in addition to the stress ratios presented in (d,e,f) for the three inferred states. This would allow to compare your inversion with other modelling studies conducted in this area (e.g. Joughin et al., 2009; Morlighem et al., 2010) and observational data. A large portion of your model domain appears white in the panels a-c indicating that observed velocities are not available here. This is not so easy to see in Rignot et al. (2011), but in Mouginot et al. (2014, Fig. 1). Please explain how do you conduct the inversion in these areas. It is not clear, how the features in the panels d-f can be explained, given the extensive data gap in a-c.

It is true that comparing the inferred fields of $|\tau_b|$ to Figs. 5-6 of Joughin et al. (2009), or to Fig. 2 of Morlighem et al. (2010), can be interresting, although colorscales are different. On the other hand, we don't think that adding maps of $|\tau_b|$ to Fig. 4 of the manuscript, which is already a heavy figure containing a lot of information, would be relevant in the context of the paper. Therefore, we have decided to add these maps of $|\tau_b|$ in a supplementary material.

As stated in the text of the manuscript, the cost function $J_v$ quantifying the misfit between modelled and observed velocities is evaluated at observation points. You are right when saying that there is a large region which is not covered by observation points. As a consequence, the solutions obtained in this region for the fields of $\alpha$ and $\gamma$ stay close to the initial guesses (i.e. $\alpha$ such that $|\tau_b| = |\tau_d|$ and $\gamma$ such that $\bar{\eta}_0 = \bar{\eta}_{0,ref}$), except that they are smoother because of the regularisation functions $J_{reg,\alpha}$ and $J_{reg,\gamma}$, which are evaluated over the whole domain. However, the lack of information in this region is not critical as the flow of ice is known to be very slow over there. We have represented, in Fig. 1 of the present document, the norm of the driving stress $|\tau_d|$ which is calculated from the gradient of the surface elevation $z_s$ as follows:

$$\tau_d = \rho_i g H \mathrm{grad}(z_s). \tag{2}$$

[Figure]

Figure 1: $|\boldsymbol{\tau}_d|$ (kPa) after initialisation.

Comparing this figure to Fig. S1 of the supplementary material shows that the fields of $|\boldsymbol{\tau}_b|$ obtained in the region where surface velocities are missing look like smoothen versions of $|\boldsymbol{\tau}_d|$. Therefore, the "wave-like" features of $|\boldsymbol{\tau}_b|/|\boldsymbol{\tau}_d|$ observed on Fig. 4d-f of the manuscript in this region come from similar features in $|\boldsymbol{\tau}_d|$, which are not present in $|\boldsymbol{\tau}_b|$. These features in $|\boldsymbol{\tau}_d|$ are likely due to the irregularity of $|\mathrm{grad}(z_s)|$ in this region, as it can be seen in Fig. 2 of the present document.

- P. 25, Fig. 5: I can't see any difference between a,b and c. The tiny little areas in between the green and grey areas appear all just red. The zoom in area should be marked in one of the figures for the whole ASE.

The first purpose of Fig. 5a-c is to show that the nodes at which $|\boldsymbol{\tau}_b|$ recalculated with the Schoof law (with $C_{max} = 0.4$) following the identification of $C_S$ differs significantly from the $|\hat{\boldsymbol{\tau}}_b|$ calculated with the linear Weertman law and used for the identification step, are very few (i.e. 8% at most). The second purpose of this figure is to show that these nodes are mostly located close to the ice shelves where ice is almost at floatation, except for some regions located far inland (bottom right corner of Fig. 5a-c) where $N$ is low because of locally very low ice thicknesses, as shown in Fig. 3 of the present document. We think that this two goals are fulfilled with the version of Fig. 5 as it was in the first version of the manuscript. It is true that differences between a,b and c are difficult to distinguish (also because these differences are actually slight), but the reader can still see some small differences between panels (e.g. west part of Thwaites Ice Shelf or upper part of PIG Ice Shelf). We could have focused on a particular region to have a better zoom in, e.g. around Thwaites Ice Shelf, but this would have been to the detriment of other regions of interrest, which would have hampered the first purpose of the figure.
The zoom in area of Figs. 5a-c has been added on Fig. 5d

- P. 27, Fig. 8: The coloured lines should be slightly thicker.

This has been done.

**References**

Borstad, C., Rignot, E., Mouginot, J., and Schodlok, M.: Creep deformation and buttressing capacity of damaged ice shelves: theory and application to Larsen C ice shelf, The Cryosphere, 7, 2013.

Brisbourne, A. M., Smith, A. M., Vaughan, D. G., King, E. C., Davies, D., Bingham, R., Smith, E., Nias, I., and Rosier, S. H.: Bed conditions of Pine Island Glacier, West Antarctica, Journal of Geophysical Research: Earth Surface, 122, 419–433, 2017.

[Figure]

Figure 2: $|\mathrm{grad}(z_s)|$ (m/100m) after initialisation.

Brondex, J., Gagliardini, O., Gillet-Chaulet, F., and Durand, G.: Sensitivity of grounding line dynamics to the choice of the friction law, Journal of Glaciology, 63, 854–866, 2017.

Budd, W., Keage, P., and Blundy, N.: Empirical studies of ice sliding, Journal of glaciology, 23, 157–170, 1979.

Cuffey, K. M. and Paterson, W. S. B.: The physics of glaciers, Academic Press, 2010.

Gillet-Chaulet, F., Gagliardini, O., Seddik, H., Nodet, M., Durand, G., Ritz, C., Zwinger, T., Greve, R., and Vaughan, D. G.: Greenland ice sheet contribution to sea-level rise from a new-generation ice-sheet model, The Cryosphere, 6, 1561–1576, 2012.

Gillet-Chaulet, F., Durand, G., Gagliardini, O., Mosbeux, C., Mouginot, J., Rémy, F., and Ritz, C.: Assimilation of surface velocities acquired between 1996 and 2010 to constrain the form of the basal friction law under Pine Island Glacier, Geophysical Research Letters, 43, 2016.

Iverson, N. R., Hooyer, T. S., and Baker, R. V.: Ring-shear studies of till deformation: Coulomb-plastic behavior and distributed strain in glacier beds, Journal of Glaciology, 44, 634–642, 1998.

Joughin, I., Tulaczyk, S., Bamber, J. L., Blankenship, D., Holt, J. W., Scambos, T., and Vaughan, D. G.: Basal conditions for Pine Island and Thwaites Glaciers, West Antarctica, determined using satellite and airborne data, Journal of Glaciology, 55, 245–257, 2009.

Larour, E., Seroussi, H., Morlighem, M., and Rignot, E.: Continental scale, high order, high spatial resolution, ice sheet modeling using the Ice Sheet System Model (ISSM), Journal of Geophysical Research: Earth Surface, 117, 2012.

Minchew, B. M., Meyer, C. R., Robel, A. A., Gudmundsson, G. H., and Simons, M.: Processes controlling the downstream evolution of ice rheology in glacier shear margins: case study on Rutford Ice Stream, West Antarctica, Journal of Glaciology, pp. 1–12, 2018.

Morlighem, M., Rignot, E., Seroussi, H., Larour, E., Ben Dhia, H., and Aubry, D.: Spatial patterns of basal drag inferred using control methods from a full-Stokes and simpler models for Pine Island Glacier, West Antarctica, Geophysical Research Letters, 37, 2010.

[Figure]

Figure 3: Ice thickness (m) used for initialisation. The black square is the zoom in area of Fig. 5a-c. The ice shelves are in green for consistency with Fig. 5 of the manuscript.

Morlighem, M., Bondzio, J., Seroussi, H., Rignot, E., Larour, E., Humbert, A., and Rebuffi, S.: Modeling of Store Gletscher's calving dynamics, West Greenland, in response to ocean thermal forcing, Geophysical Research Letters, pp. n/a–n/a, https://doi.org/10.1002/2016GL067695, URL http://dx.doi.org/10.1002/2016GL067695, 2016GL067695, 2016.

Paterson, W. and Budd, W.: Flow parameters for ice sheet modeling, Cold Regions Science and Technology, 6, 175–177, 1982.

Payne, A., Huybrechts, P., Abe-Ouchi, A., Calov, R., Fastook, J., Greve, R., Marshall, S., Marsiat, I., Ritz, C., Tarasov, L., et al.: Results from the EISMINT model intercomparison: the effects of thermomechanical coupling, Journal of Glaciology, 46, 227–238, 2000.

Schoof, C.: The effect of cavitation on glacier sliding, Proceedings of the Royal Society A: Mathematical, Physical and Engineering Science, 461, 609–627, 2005.

Seroussi, H., Nakayama, Y., Larour, E., Menemenlis, D., Morlighem, M., Rignot, E., and Khazendar, A.: Continued retreat of Thwaites Glacier, West Antarctica, controlled by bed topography and ocean circulation, Geophysical Research Letters, pp. n/a–n/a, https://doi.org/10.1002/2017GL072910, URL http://dx.doi.org/10.1002/2017GL072910, 2017GL072910, 2017.

Smith, A. M., Jordan, T. A., Ferraccioli, F., and Bingham, R. G.: Influence of subglacial conditions on ice stream dynamics: seismic and potential field data from Pine Island Glacier, West Antarctica, Journal of Geophysical Research: Solid Earth, 118, 1471–1482, 2013.

Tsai, V. C., Stewart, A. L., and Thompson, A. F.: Marine ice-sheet profiles and stability under Coulomb basal conditions, Journal of Glaciology, 61, 205–215, 2015.

Van Liefferinge, B. and Pattyn, F.: Using ice-flow models to evaluate potential sites of million year-old ice in Antarctica, Climate of the Past, 9, 2335, 2013.

Weertman, J.: On the sliding of glaciers, Journal of glaciology, 3, 33–38, 1957.

Winkelmann, R., Martin, M. A., Haseloff, M., Albrecht, T., Bueler, E., Khroulev, C., and Levermann, A.: The Potsdam parallel ice sheet model (PISM-PIK)-Part 1: Model description, The Cryosphere, 5, 715, 2011.

Yu, H., Rignot, E., Morlighem, M., and Seroussi, H.: Full-Stokes modeling of grounding line dynamics, ice melt and iceberg calving for Thwaites Glacier, West Antarctica, The Cryosphere Discussions, 2016, 1–24, https://doi.org/10.5194/tc-2016-101, URL `http://www.the-cryosphere-discuss.net/tc-2016-101/`, 2016.